# On the Tool Manipulation Capability of Open-source Large Language Models

## Abstract

Recent studies on software tool manipulation with large language models (LLMs) mostly rely on closed model APIs. The industrial adoption of these models is substantially constrained due to the security and robustness risks in exposing information to closed LLM API services. In this paper, we ask *can we enhance open-source LLMs to be competitive to leading closed LLM APIs in tool manipulation, with practical amount of human supervision.* By analyzing common tool manipulation failures, we first demonstrate that open-source LLMs may require training with usage examples, in-context demonstration and generation style regulation to resolve failures. These insights motivate us to revisit classical methods in LLM literature, and demonstrate that we can adapt them as model alignment with programmatic data generation, system prompts and in-context demonstration retrievers to enhance open-source LLMs for tool manipulation. To evaluate these techniques, we create *ToolBench*[1], a tool manipulation benchmark consisting of diverse software tools for real-world tasks. We demonstrate that our techniques can boost leading open-source LLMs by up to 90% success rate, showing capabilities competitive to OpenAI GPT-4 in 4 out of 8 ToolBench tasks. We show that such enhancement typically requires about one developer day to curate data for each tool, rendering a recipe with practical amount of human supervision.

## 1 Introduction

Tool-augmented large language models (LLMs) recently emerge as a research frontier. Such augmented LLMs demonstrate tool manipulation capabilities which automate software operations through natural language instructions [1, 2, 3, 4, 5]. Despite the fact that open-source LLMs greatly shrink the quality gap towards proprietary closed LLMs in tasks such as chatbot [6, 7, 8, 9], recent tool-augmented LLMs still mostly rely on closed LLM APIs [1, 2, 3, 4]. This leads to a fundamental barrier for the industrial adoption of these augmented LLMs due to security and robustness risks associated with exposing enterprise-internal workflows and information to closed LLM APIs [10, 11]. To this end, we ask *can we build on open-source LLMs with practical amount of human supervision and achieve tool manipulation capabilities competitive to closed LLMs.*

Table 1: Example of tool manipulation errors. Errors are highlighted in red.

| | |
|---|---|
| **Goal** | # To move the robot to position (x, y)
robot.move_to(x, y)
# To raise the arm by a given height
robot.raise_arm(height)
Task: how to move a robot to (20, 30)? |
| **Expected results** | robot.move_to(20, 30) |
| **Wrong API** | robot.**raise_arm**(20) |
| **Wrong Arguments** | robot.move_to(**30, 20**) |
| **Non-executable** | **You can create a robot with**
**robot = Robot()**
**and move it to the target location by**
robot.move_to(20, 30) |

In this paper, we first demystify key challenges for tool manipulation using open-source LLMs; we then leverage the insights to suggest practical recipes for enhancement. Concretely, we study the setting shown in Figure 1 where LLMs take in a natural language instruction as the goal and generate API calls to accomplish the goal. Although we expect a quality gap between the open-source and closed LLMs [12], what we observe is a far more severe disparity. Specifically, for an on-sale house searching tool, a leading open LLM for code generation fails every test case while the OpenAI GPT-4 [13] attains 77% success rate across

---

[1]Available at https://github.com/neurips2023-userXXX/ToolBench

Figure 1: Tool manipulation setup. We augment LLMs as action generators with access to API documentations. In the single-step scenario, an action generator directly generates API calls to accomplish the goal. A multi-step action generator further iterates with environment feedback to generates the next-step API calls until hits an exit state.

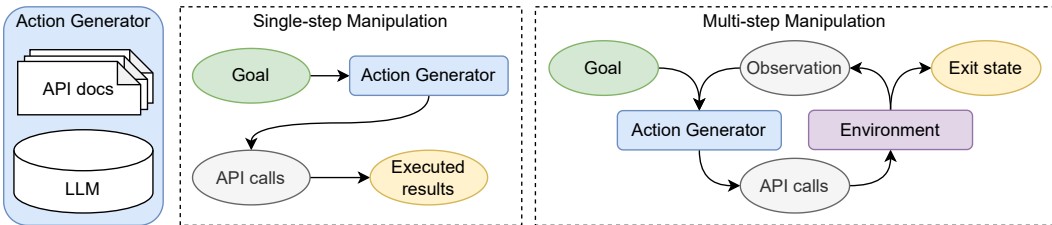

the same one hundred examples. This observation motivates us to study the challenges for open-source LLMs to attain strong tool manipulation capability.

During our investigation, we identify three key challenges listed in Table 1 that impede the performance of open-source LLMs in tool manipulation. Firstly, open-source models often struggle to accurately identify API names, whereas closed LLMs demonstrate the capability to invoke the correct APIs without explicit usage examples or documentation during inference. Secondly, without demonstration examples, open-source LLMs often fail to populate the appropriate values for API arguments. Thirdly, we demonstrate that open-source LLMs tend to produce non-executable generation, such as natural language beyond the desired code.

Our insights suggest us to revisit three *simple* techniques from LLMs for conventional NLP tasks. In the context of tool manipulation, we adapt them with practical amount of supervision and use them to enhance open-source LLMs. *Model alignment:* To first internalize API usage knowledge, we perform instruction tuning [14, 15] with programatically generated data. Specifically, we first write a few dozens of templates on goals and corresponding API calls. We then pragmatically bootstrap the data volume by instantiating templates with concrete key word values. *In-context demonstration retriever:* Inspired by retrieval-augmented generation [16, 17, 18], we additionally enhance the LLMs with a *retriever* to leverage in-context demonstrations during inference. This module selects demonstration examples with the most semantically similar goals from a human-curated pool of examples. *System prompt:* Finally we embed goal descriptions into a pre-defined system prompt which provides inference-time guidelines to generate executable API calls; such system prompts were shown to regulate language style in chatbots [19]. We show that these three techniques only require a small amount of human supervision.

To extensively evaluate the inspired techniques, we present *ToolBench*, a benchmark suite on eight diverse tools ranging from Google Sheets manipulation to controlling robots [20]. It enables the first publicly-available quantitative evaluation test bench among the ones brought up in the tool-augmented LLM literature [2, 3]. For the software tools in our benchmark, LLMs need to accomplish a variety of goals by selecting and combining API functions from up to a hundred candidates.

Using the tools in the ToolBench suite, we first empirically show that leading open-source LLMs can demonstrate up to 78% lower success rate when compared to the OpenAI GPT-4 APIs. We then demonstrate that these simple techniques can substantially improve the success rate of open-source LLMs by up to 90%, attaining results competitive or better than OpenAI GPT-4 models in 4 out of the 8 tools in our benchmark[2]. To reveal the impact of different techniques, we provide evidence that aligning model with synthetic data primarily contributes to the significant improvement of open-source LLMs. The system prompt and the in-context demonstration retriever further enhance the performance. During the enhancement process, we observe that, on average, it takes just one day for a developer to craft the in-context demonstrations and curate the templates for generating model alignment data. This implies that the recipe requires a practical level of human supervision.

---

[2]We apply the same system prompt and in-context example retriever for GPT-4. Model alignment is not applicable to GPT-4 as there is no publicly available tuning APIs for it during our experiments.

## 2 Background

In this paper, we study the scenario where software users intend to translate a natural language goal description $g$ into a sequence of application programming interface (API) calls $C_g = \{c_0, c_1, \cdots, c_{n_g}\}$ to accomplish the goal. We study tool manipulation with open-source LLMs in this specific setting, because APIs serve as the prevalent abstraction for developers and users in modern software systems.

**Large language model**    Autoregressive language models encode probabilities of the next word $x_{N+1}$ given $x_0, x_1, \cdots, x_N$ as the context sequence [21]. By sampling from this conditional probability $p(x_{N+1}|x_0, x_1, \cdots, x_N)$ iteratively, it generates language continuations from given contexts. In the recent wave of scaling up model size and training data volume, transformer-based language models show unprecedented capability in instruction following for text and code generation [22, 23, 24]. In the context of tool manipulation, we cast goal descriptions and optional information as an instruction in the context and task the LLMs to generate code for API calls as the continuation.

**Action generator**    A key implementation for tool manipulation is an action generator $\mathcal{A}$ which maps a goal $g$ to a series of API calls $C_g$ to accomplish that goal. As open-source LLMs likely have not seen the information regarding the relevant APIs, we augment an LLM $\mathcal{M}$ into an action generator by providing access to a pool of $m$ candidate API functions $\mathcal{D} = \{d_0, d_1, \cdots, d_m\}$. Due

**Algorithm 1** API Call Generation

**Input:** Goal $g$, API docs $\mathcal{D}$, action generator $\mathcal{A}$
**Input:** Optional info $O$
1: **procedure** ACTIONGEN($g$, $\mathcal{D}$, $\mathcal{A}$, $O$)
2:     $\mathcal{D}_g \leftarrow \mathcal{R}(g, \mathcal{D})$    ▷ Retrieve API functions
3:     $C_g \leftarrow \mathcal{A}(g, \mathcal{D}_g, O)$    ▷ API call generation
4:     **return** $C_g$
5: **end procedure**

to the input sequence length limit of LLMs, we provide an optional retriever $\mathcal{R}$ to retain a relevant subset of API documents $\mathcal{D}_g = \mathcal{R}(g, \mathcal{D}) \in \mathcal{D}$. Thus, the action generator produces the sequence of API calls $C_g = \mathcal{A}(g, \mathcal{D}_g, O)$, where $O$ represents the optional information that can be included in the prompt. This is a naive way of retrieval augmented generation [18, 25, 26] and we employ an off-the-shelf retriever implementation [27] for our study, but we also highly encourage the community to explore algorithms tailored for the action generator.

**Single and multi-step tool manipulation**    As shown in Figure 1, an action generator may interact with software in either a single-step or a multi-step scenario. In a single-step scenario, action generator directly produces an API call sequence $C_g = \mathcal{A}(g, \mathcal{D}_g, \emptyset)$. In a multi-step scenario, the action generator produces a series of API calls $C_{g,i}$ at $i^{th}$ iteration, where $C_{g,i}$ is used to interact with a predefined environment $\mathcal{E}$ and generates the observation $O_i = \mathcal{E}(C_{g,i})$. The observation is then used to generate $C_{g,i+1} = \mathcal{A}(g, \mathcal{D}_g, O_i)$. The process stops at an exit state $S_g$. During evaluation, the execution results of $C_g$ from the single-step scenario and the $S_g$ from the multi-step version are compared against the ground-truth label. Note that, the main difference between the two scenarios lies in whether or not to interact with environment $\mathcal{E}$; while both $C_g$ and $C_{g,i}$ can individually be a series of more than one API calls to accomplish a given task.

## 3 Challenges for open-source LLMs

To demystify key challenges, we study the behaviors of open-source LLMs in tool manipulation. By analyzing common mistakes in a weather query task, we discover three challenges to attain strong tool manipulation capabilities. As shown in Table 1, we observe that open-source LLMs often face difficulty in (1) API selection, (2) API argument population, and (3) generat-

Table 2: Categorized typical tool manipulation error types on a weather query tool.

|  | GPT-4 | LLaMA | StarCoder | CodeGen |
|---|---|---|---|---|
| Failure rate | 19% | 61% | 68% | 93% |
| API selection | 0% | 22% | 22% | 30% |
| Args. populating | 14% | 32% | 23% | 63% |
| Non-executable | 5% | 7% | 23% | 0% |

ing legitimate and executable code [3]. These insights are described in detail in this section and inspire the techniques to alleviate the challenges in Section 4.

---

[3]If a failure case has multiple errors, we categorize it by the first triggered category in the following order: non-executable generation, wrong API selection, wrong argument populating

**Difficulty in API selection** We observe that API selection failures often involve using incorrect APIs and even hallucinating non-existent API names. To quantitatively understand the intrinsic capability in API selection, we compare open-source LLMs to GPT-4 without providing any documentation or in-context demonstrations during inference. The results, as shown in Figure 2 for the weather query tool OpenWeather, reveal that GPT-4 can choose the right API without additional information beyond the goal, while open-source models struggle. Such capability disparity entails that *closed LLMs potentially internalize knowledge of API usage during training.*

Figure 2: On calling OpenWeather APIs, (left) without any API documentation exposure during inference, closed LLMs attain high accuracy in selecting APIs , implying potential example usage exposure during training. However, (right) hand-picked oracle one-shot demonstration improves success rate over zero-shot, showing the roofline impact of in-context demonstrations.

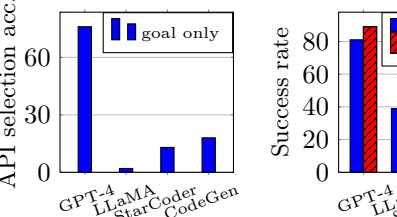

**Confusion in populating arguments** After the action generator selects the appropriate APIs, the subsequent challenge lies in parsing the goal description and populating the API arguments. At this stage, we observe that open-source models often provide wrong values for the required API arguments. The confusion in argument populating contributes to up to 63% of the failures in open-source models, as shown in Table 2. In an attempt to mitigate this issue, we provide the LLMs with a hand-picked oracle in-context demonstration which achieves the same goal with different argument values. We show in Figure 2 that the hand-picked oracle examples improve success rates by up to 45%. It is important to note that oracle examples are not intended as a solution for argument populating confusion, as they are hand-picked on a per-test-case basis. Nonetheless, these observations suggest that *in-context demonstrations can substantially enhance open-source LLMs for tool manipulation.*

**Non-executable generation** The third common failure of open-source LLMs is non-executable generation. Such failures encompass issues such as language verbosity around API calls and adherence to natural language based guidelines, as shown in Table 1. Open-source models exhibit such errors in 23% of one hundred weather query cases. These observations underscore *the necessity of regulating open-source LLMs to exclusively generate code.*

## 4  BOOSTING OPEN-SOURCE LLMS FOR TOOL MANIPULATION

The insights from Section 3 emphasize the importance of tuning with API usage examples, in-context demonstration and generation regulation in the domain of tool manipulation. In this section, *we revisit three techniques from the LLM literature and adapt them to address the aforementioned challenges,* **using a practical amount of human supervision.** We first introduce model alignment with programatically curated data to internalize API usage knowledge in Section 4.1. We then discuss augmenting open-source LLMs with an in-context demonstration retriever in Section 4.2. Lastly, we apply a system prompt to regulate generation in Section 4.3. These techniques collectively serve as a strong baseline for alleviating the challenges presented in Section 3 and inspiring further innovations.

### 4.1  MULTI-TOOL MODEL ALIGNMENT WITH PROGRAMMATIC DATA CURATION

Model alignment, through tuning LLMs with usage examples, plays a vital role in improving LLMs for capabilities such as instruction following and conversation [14, 19, 28]. In light of our insights from in Section 3, we recognize the potential of model alignment with API usage examples to improve API selection capability. To practically leverage such alignment for tool manipulation, it requires a data curation strategy without massive manual example writing. Towards this end, we prototype a method which generates usage examples from human-curated templates.

Figure 3 depicts our flow to generate alignment data. We create a handful of templates consisting of goal descriptions and corresponding API calls. These templates contain one or more placeholder pairs. Each of these pairs maps to a key word in the goal and an

argument in the corresponding API calls. We also provide a pool of candidate values for each keyword and randomly choose values to fill in the placeholders within the template.

Given a tool with $n$ candidate APIs, we only require $\mathcal{O}(n)$ human-curated templates to ensure practical human supervision. Specifically we use a principle where each of the $n$ APIs is encouraged to appear in at least one template. In practice, we find it takes on average one day for one developer to curate the data for one software tool in our benchmark; this includes writing the goal templates, providing the pool of argument values and generate the data. We provide example templates we use for different tools in Appendix C. With data curated for all the tools, we perform model alignment tuning *jointly for all tools and produce a single model.*

Figure 3: Programmatic training data generation using templates and random values

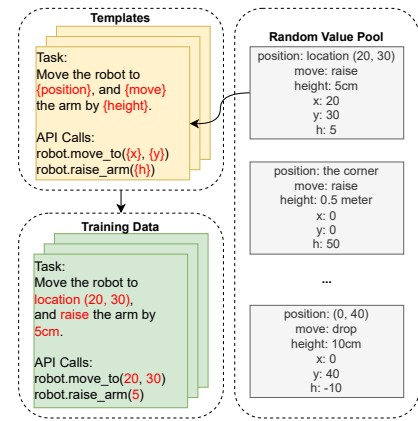

## 4.2 Demonstration retrieval

In Section 3, we demonstrate the efficacy of hand-picked oracle examples in improving argument populating. However, extending from oracles to practical in-context demonstration poses two challenges. First, given $n$ API function candidates, there are exponentially many combinations of API calls associated with different goals. Thus, LLMs should be capable of generalizing to a wide variety of goals based on a limited number of examples. Second, to ensure effective demonstration, it is important to provide LLMs with only the relevant examples without human interventions. To fulfill the above two desiderata, we augment open-source LLMs with a demonstration retriever module[4]. During action generation, top-k $(\hat{g}, \hat{C}_g)$ pairs that are the closest to a given $g$, are retrieved from the demonstration examples pool and prepended in the prompt for few-shot in-context learning. We also make sure the size of the demonstration examples pool grows linearly with the total number of API functions.

**Validation** To verify the effectiveness of demonstration examples in practice, we empirically show that the retrieved demonstrations can improve the success rate on goals requiring API combinations unseen in the example repository. In particular, we evaluate this approach on the home search task which exposes 15 API functions and requires multiple functions to accomplish each goal. With only 10 human-curated demonstrations that do not precisely match any of the 100 test cases in terms of API combinations, the retrieved demonstrations can boost the success rate by up to 79% across open-source LLMs and make GPT-4 nearly perfect, as shown in Figure 4. This shows that the demonstration examples can improve tool manipulation for unseen types of goals with a repository of size $\mathcal{O}(n)$ only.

Figure 4: In-context demonstration can improve both closed and open-source models on Home Search, a tool for browsing houses on sale.

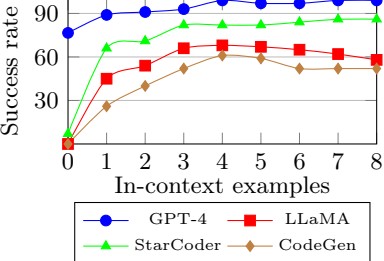

## 4.3 Generation regulation with system prompts

The use of system prompts is a well-established technique in chatbots powered by LLMs [19]. By incorporating into human-chatbot conversations, system prompts can effectively control the natural language style of the generated responses. In the context of tool manipulation, our system prompt first defines a format that combines text sections containing goals, retrieved API functions, and demonstration examples. It then provides explicit guidelines in natural language, instructing the LLMs to generate code exclusively. The exact system prompt is presented in Appendix B. Constructing the system prompt is a one-time effort for each task, which requires minimum human development effort.

---

[4] We used the BM25 retriever in all our experiments

Table 3: Tasks in the ToolBench. We provide demonstration examples for few-shot in-context-learning while test cases are for quantitatively evaluation. We develop API complexity, a metric to quantify the challenge level in generalizing to unseen API combinations; higher complexity indicates more challenging tasks. We package the challenges beyond API complexity as advanced reasoning. We refer to Appendix A for more details on these tasks.

| Task | Single Step | | | | | | Multi-Step | |
|---|---|---|---|---|---|---|---|---|
| | Open Weather | The Cat API | Home Search | Trip Booking | Google Sheets | VirtualHome | WebShop Long / Short | Tabletop |
| *Data* | | | | | | | | |
| API functions | 9 | 6 | 15 | 20 | 108 | 40 | 2 | 32 |
| Demonstration examples | 18 | 12 | 10 | 11 | 10 | 83 | 1533 / 200 | 74 |
| Test cases | 100 | 100 | 100 | 120 | 70 | 100 | 100 | 105 |
| *Level of challenges* | | | | | | | | |
| API complexity | 2.2 | 1.4 | 7.3 | 11.1 | 8.4 | 12.3 | 0.0 | 4.6 |
| Advanced reasoning | | | | | ✓ | | ✓ | ✓ |

# 5 TOOLBENCH: A NEW TOOL MANIPULATION BENCHMARK

To evaluate open-source LLMs in the domain of tool manipulation, we curate a benchmark suite from both existing datasets and newly collected ones. This benchmark stands out as the first open-source test bench with predefined test cases for quantitative evaluation, distinguishing it from recent tool manipulation research using closed LLMs [2, 3]. In this section, we introduce the software tools and the evaluation infrastructure. We also demonstrate the level of challenges posed by each tool, in terms of the ability to generalize to unseen API combinations and the requirement for advanced reasoning.

## 5.1 SOFTWARE TOOLS AND EVALUATION INFRASTRUCTURE

As shown in Table 3, our benchmark consists of five tasks we collected and three tasks derived from existing datasets, including VirtualHome[29, 30], Webshop[31] and Tabletop[20]. They cover both single-step and multiple-step action generation, which requires selecting and combining from 2 to 108 API functions to accomplish the goals. Each task consists of approximately 100 test cases, including goal descriptions and the ground truth API calls. We also provide a limited number of demonstration examples to aid model predictions[5]. We include a comprehensive introduction and analysis of each task within the benchmark in Appendix A.

We use *success rate* as the primary evaluation metric for most tasks, except for the WebShop where we report rewards, as well as for VirtualHome where we use executability and Longest Common Subsequence (LCS), following the original metrics proposed by the respective authors. To facilitate evaluation, we build an infrastructure that executes the API calls generated by the action generators and assess the final outcome. This process enables reliable evaluation of tool manipulation capabilities without restricting the action generators to perfectly match the ground truth API calls.

## 5.2 LEVEL OF CHALLENGES

To assess the level of challenge, we examine ToolBench tasks based on their API complexity and the requirement for advanced reasoning. Intuitively, API complexity indicates the challenges in generalizing to unseen API combinations and non-default argument values. Challenges beyond API complexity then involve advanced reasoning.

**API Complexity** To quantify the challenge in generalizing to unseen API combinations, we develop a task-agnostic complexity score $S \in \mathbb{R}_0^+$, where $S(\mathcal{T}, \mathcal{X}, \mathcal{D}) = \mathbb{E}_{t \in \mathcal{T}} \min_{e \in \mathcal{X}} d(t, e)$. It averages over all the test samples in the test set $\mathcal{T}$ on the minimum distance between $t$ and any demonstration example $e$ from the example pool $\mathcal{X}$. In particular, the distance $d(t, e)$ between each test sample $t$ and a demonstration example $e$ is negatively proportional

---

[5]For WebShop, we find that more than $\mathcal{O}(n)$ demonstration examples can improve the success rate. Nonetheless, these examples can be acquired from programmatic software operations without heavy human curation.

to the probability of transforming the API combination of $e$ to match that of $t$, by randomly dropping the API functions irrelevant to $t$ and inserting the uncovered API functions required by $t$ from the API pool $\mathcal{D}$. We refer to the details of the complexity score to Appendix D and list their values in Table 3. The score is non-negative and the higher the score is, the more complex a task is. Despite the fact that this complexity score reflects the challenge level of API selection, it does not capture all the difficulties of a task. A task with low complexity score can still be very challenging as it might require advanced reasoning. For instance, even though Webshop is challenging, the API selection complexity of it is zero. This is because there are only two API functions requiring only one argument each in Webshop, and they are both covered by the examples, so there is no API selection complexity.

**Advanced reasoning** Within our benchmark, advanced reasoning encompasses challenges beyond generalizing to unseen API combinations. These challenges include non API-based coding for tasks such as Google Sheets and Tabletop, as well as decision-making based on observations returned from the WebShop environment. For instance, in the Google Sheets example shown in Table 4, the coordinate of the beef price's cell ("C2") cannot be easily derived from either the goal or the table itself. The action generator needs to understand the content or write additional python code to derive this coordinate before calling the API function. In the similar scenario, WebShop task requires the action generator to extract the exact button ID to click on the webpage given the description. These challenges, categorized as advanced reasoning, complement the API complexity category.

Table 4: A typical task of Google Sheets manipulation. It requires advanced reasoning on populating correct arguments.

| Product | Cost | Price |
|---------|------|-------|
| beef | 1 | 3 |
| pork | 5 | 4 |
| chicken | 10 | 11 |

Task: Update beef's price to 10.
Action:
**worksheet.update("C2", 10)**

## 6 EXPERIMENT

In this section, we leverage the ToolBench to empirically validate the techniques introduced in Section 4. First, to concretize the capability gap between open-source and closed LLMs, we demonstrate that OpenAI GPT-4 API can have substantially higher success rate than representative open-source LLMs in Section 6.2. We then show in Section 6.3 that the simple techniques in Section 4 can boost open-source LLMs to achieve success rates competitive to in-context-learning with GPT-4 APIs[6] in four out of the eight tasks. Through ablation studies in Section 6.4, we additionally show that model alignment does the heavy lifting for boosting open-source LLMs, while system prompt and in-context learning robustify LLMs for further improvement.

### 6.1 EXPERIMENT SETUP

To establish strong baselines, we use GPT-4 API as the representative closed LLM in our study because it attains the leading accuracy in mainstream NLP tasks. In our study, we compare LLAMA-30B [32], StarCoder [33] and CodeGen-16B-mono [34] to GPT-4. LLAMA represents open research models, while StarCoder and CodeGen are publicly available for both research and commercial purposes. We choose these three models due to their superior performance on ToolBench among open-source models as shown in Table 8[7]. In our experiments, we consider the zero-shot setting as the out-of-the-box configuration where only API documentation is provided without any demonstration examples. We use this configuration to understand the initial gap in capabilities among models. We then incorporate all available techniques on top of this initial configuration to assess their benefits. For the original Tabletop dataset [20], which includes examples in a few-shot setting without explicit API definitions, we only evaluate settings with in-context demonstrations. More detailed setup information is included in Appendix C. We run each job 3 times with different random seeds and report average accuracy. The variation is minimal, so we ignore them in the main paper but report them in appendix.

---

[6]GPT-4 tuning APIs were not released by the time this work is done.

[7]Surprisingly, we observe that for tool manipulations, open-source LLMs instruction-tuned for conventional NLP tasks do not outperform their base models before tuning.

Table 5: Capability gap in tool manipulation is substantial between closed API and open-source LLMs in the out-of-the-box zero-shot setting. Using model alignment, the in-context demonstration retriever and the system prompt, open-soured LLMs attain significant boost in success rate. GPT-4 is enhanced with the retriever and system prompt. Tabletop is only evaluated in the few-shot fashion.

| Task | Open Weather | The Cat API | Home Search | Trip Booking | Google Sheets | VirtualHome | WebShop Long | WebShop Short | Tabletop |
|---|---|---|---|---|---|---|---|---|---|
| _Zero-shot Baseline_ | | | | | | | | | |
| GPT-4 | 81.3 | 97.4 | 76.6 | 91.5 | 5.7 | 40.8 / 8.0 | 0.0 | | - |
| LLaMA-30b | 39.0 | 49.0 | 0.0 | 0.0 | 0.0 | 78.0 / 0.3 | 0.0 | | - |
| StarCoder | 32.0 | 71.0 | 7.0 | 13.3 | 5.9 | 22.0 / 3.7 | 0.0 | | - |
| CodeGen-16B-mono | 7.0 | 78.0 | 0.0 | 0.0 | 1.4 | 4.0/ 1.0 | 0.0 | | - |
| _Enhanced w/ techniques_ | | | | | | | | | |
| GPT-4 | 99.0 | 98.0 | 98.0 | 99.2 | 68.6 | 29.0 / 21.7 | 0.0 | 0.0 | 83.8 |
| LLaMA-30b | 100.0 | 94.0 | 87.0 | 85.8 | 2.9 | 16.0 / 24.3 | 0.0 | 0.0 | 7.5 |
| StarCoder | 99.0 | 97.0 | 83.0 | 80.8 | 21.2 | 31.0 / 18.4 | 0.0 | 0.0 | 13.9 |
| CodeGen-16B-mono | 97.7 | 99.0 | 82.0 | 77.5 | 19.8 | 29.0 / 17.2 | 0.0 | 3.5 | 16.2 |

## 6.2 Capability Gap

Table 5 exhibits significant disparities in tool manipulation between the closed GPT-4 API and open-source models in the out-of-the-box zero-shot setting. For simpler tasks, namely Open Weather and the Cat API, which require only one API call for each goal, the open-source models exhibit success rates up to 74% lower than GPT-4. Furthermore, on all the remaining tasks other than the Webshop, none of the LLAMA, the StarCoder and the CodeGen model can reach meaningful accuracy or compare with GPT-4. These results highlight an opportunity to enhance open-source LLMs.

## 6.3 Boosting open-source LLMs

To boost the open-source LLMs, we first perform model alignment using programmatially generated data. We then apply a system prompt and a 3-shot demonstration retriever during inference. Given GPT-4 does not provide tuning APIs, we enhance the out-of-the-box GPT-4 with the same system prompt and demonstration retriever as the baseline. The improvements from the combined enhancement techniques are shown in Table 5, where the success rates of the open-source LLMs can improve up to 90%. As a result, the open-source models achieve competitive or better success rates on 4 out of 8 tasks, including Open Weather, the Cat API, VirtualHome and WebShop. Moreover, on Home Search and Trip Booking, the gap between the LLAMA model and the GPT-4 API is reduced to 11% and 13.4% respectively, compared to the initial gap of up to 91%. Despite the fact that open-source models are still lagging behind on the Google Sheets and Tabletop, these observations show that _our recipe can significantly improve the performance of open-source LLMs and attain success rates comparable to GPT-4 API on many of the ToolBench tasks._

**Human supervision** To identify the practicality of an enhancement recipe, the amount of required human supervision is a crucial factor. In our approach, human supervision is primarily in the form of in-context demonstration examples and alignment data templates. Regarding the demonstration examples, we provide 10 to 83 examples for each task as shown in Table 3, except for WebShop given its difficulty in advanced reasoning. As shown in Table 9, the number of templates for alignment data is typically less than 100 for each task. We observe that providing these supervisions takes one developer day on average, making it practical in terms of the time cost on human supervision.

**Remaining challenges** We observe that the boosted open-source LLMs still have relatively low success rates on tasks that require advanced reasoning, such as Google Sheets, WebShop and Tabletop tasks. This implies the need to further enhance the reasoning capabilities of open-source models. We are excited to further improve the model quality with the community to address those challenges. However, we can point out some potential directions based on our comprehensive benchmark of different open-source models on the ToolBench in Appendix B Table 8. We found that larger models always have better performance on tool using when the training data is fixed and code generation capability directly correlates with tool using.

## 6.4 Ablation Study

We break down the contribution of the techniques in two ways. First, we apply each technique individually on top of the out-of-the-box zero-shot configuration and evaluate its impact. As shown in Table 6, both the 3-shot in-context demonstration and model alignment techniques bump up the success rates across all tasks, while the system prompt only benefits simple tasks that involve relatively fewer API calls for each goal.

Next, we consider the combination of all techniques and remove them one at a time to evaluate their relative contributions within the full system. As shown in in Table 6, solely removing model alignment triggers success rate degradation in up to 7

Table 6: The number of ToolBench tasks improved (+N) or hurt (-N) over the baselines when adding or dropping techniques.

|  | LLaMA | StarCoder | CodeGen |
|---|---|---|---|
| **Zero-shot** | - | - | - |
| + Sys. Prompt | +4 | +4 | +4 |
| + 3-shot | +8 | +8 | +8 |
| + Alignment | +7 | +7 | +7 |
| **Full system** | - | - | - |
| - Sys. Prompt | -0 | -2 | -3 |
| - 3-shot | -3 | -4 | -5 |
| - Alignment | -5 | -5 | -7 |

tasks, while removing either in-context demonstration up to 5 tasks and dropping system prompt up to 3. We notice that the tasks that are not significantly impacted when removing techniques are typically the ones with relatively low success rate (usually <20% even in the full system). Thus, those accuracy changes are hypothetically subject to high variance and fluctuation. The full results from the experiments in this section can be found in Table 11.

## 7 Related work

Our work establishes a strong connection to the LLM-driven program synthesis. In contrast to the conventional rule-based code generation in popular compilation frameworks [35], recent auto-regressive LLMs such as CodeGen[34], SantaCoder[36] and StarCoder[33] treat the problem as a sequence generation task and demonstrate superior capabilities in emitting semantically correct computer programs. We use CodeGen as a representative from these models in our study for API call generation.

Tool manipulation are also known as tool augmented learning [3, 37]. Some of the works seek to augment generations with the execution results from various tools[1, 38, 39, 26, 40, 41, 42], while another line of works focus on executing the tools themselves, including embodied robotic learning [20, 30, 43, 44, 45], and automation for other tools [31, 46, 47, 48]. We focus on the study of the second stream with different models and techniques.

Recent works in tool manipulation with LLMs mostly study techniques to enhance in-context-learning with closed LLMs APIs [1, 2, 3, 4, 5]. In contrast, we study simple techniques to allow for developers to practically build on top of open-source LLMs. The three techniques we mention in this paper [19, 22, 26, 49] are well studied in the conventional NLP tasks. We revisit and adapt them in the context of tool manipulation on open-source models with a practical amount of human supervision. In the recent LLM literature, there are several works presenting tool manipulation benchmarks [2, 3]. Compared to these benchmarks, the ToolBench is the first one providing predefined test cases to evaluate real execution results.

## 8 Conclusion

In this paper, we answer the question *can we enhance open-source LLMs to compete with leading closed LLM APIs in tool manipulation, with practical amount of human supervision.* Drawing from our observations of the common tool manipulation failures and insights from the literature on conventional NLP tasks with LLM, we propose to instantiate model alignment with programmatical data generation, system prompts, and in-context demonstration retrievers to improve the tool manipulation capability of open-source models. To comprehensively evaluate the impact of these techniques, we create the *ToolBench*, a benchmark consisting of diverse software tools for real-world tasks. Our results demonstrate that these techniques can make the leading open-source LLMs competitive with the OpenAI GPT-4 in 4 out of 8 ToolBench tasks, all achieved with a practical amount of human labeling effort.

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
