In the appendix section, we provide detailed information on the following aspects of our study. In Appendix A, we present the background and curation details for the 8 tasks included in ToolBench. Appendix B focuses on the performance evaluation of an extensive suite of LLMs on ToolBench. In Appendix C, we delve into the details of model alignment, including the process of generating the training data and training details. We also provided the full spectrum of results for the experiments in Section 6. Finally, in Appendix D, we introduce the API selection complexity score system, and demonstrate its effectiveness and implication in measuring task complexity.

## A   BENCHMARK DETAILS

### A.1   OPENWEATHER

This task involves using the REST API to interact with OpenWeather website[8]. We include 9 types of API calls that cater to 9 categories of queries, including but not limited to retrieving current weather data in a city, obtaining air quality data at a specific longitude and latitude, and acquiring weather forecast data for a location specified by a zip code. Making each type of API calls involves correctly filling 2 to 3 required parameters (such as `lon` for longitude and `lat` for latitude) and 0 to 3 optional parameters (such as `lang` for language and `units` for units of measurement), depending on the requirements specified in each query. In total, we develop 100 unique queries for the 9 categories and 2 demonstration examples for each category. To assess the quality of the LLM's generation, we look for the first line beginning with the word "curl", if it exists. We then execute this line using the shell process. If the shell process returns a non-zero value, we declare "not executable" for this generation. On the other hand, if the code can be executed, we compare the returned response with the corresponding result from the ground-truth Curl request. The model's generation will be considered successful if the output matches the expected result precisely.

### A.2   THE CAT API

This task is a similar REST API task as the OpenWeather, but it involves making all the `GET`, `DELETE`, or `POST` request to The Cat API website[9]. There are 6 types of API calls for 6 types of queries, including deleting a cat image from the user's list of favorites, adding an image to the user's list of favorites, returning the list of favorite images, voting up or down to an image, and searching for cat images with filtering requirements. We develop 100 queries for the test set and 2 demonstration examples for each category. To evaluate the executability and success of the LLM's generation in these scenarios, we follow a similar procedure as that of the Open Weather task. It is worth noting that for queries related to removing an image from the list of favorites, we compare the LLM's generation verbatim with the ground-truth label since duplicated deletion would inevitably lead to failure if executed.

### A.3   HOME SEARCH

This task is designed to replicate the process of searching for homes at a specific location based on certain criteria. We design the API with 15 functions, including

- `set_location` which sets the desired location;
- `set_but_or_rent` which specifies whether the user is looking to buy or rent a home;
- 12 functions for setting criteria, such as home prices, number of bedrooms, and home square footage;
- `search` which submits the criteria to get search results.

We consider executability and f1 score of the generated action. To ensure executable searches, the agent should make a sequence of function calls that starts with `set_location` and `set_buy_or_rent`, followed by the criterion-setting functions, and then ends with a call to the `search` function. If executable, an f1 score is computed between the criteria set by the

---

[8]https://openweathermap.org/api
[9]https://thecatapi.com

generated program and that by the ground-truth program. We develop a test set consisting of 100 queries that asked for home options with varying criteria combinations and provide 10 demonstration examples. To test the LLM's ability to utilize unseen API functions, we intentionally exclude 3 criterion-setting functions from all demonstration examples.

### A.4 TRIP BOOKING

The Trip Booking task is similar to the Home Search task but with more advanced dependency requirements among function calls. It simulates the process of submitting search requests for transportation tickets, hotel rooms, or both based on specific requirements like locations, dates, and the number of tickets required. We design 20 functions for the three types of booking scenarios. Depending on the scenario, some function calls may be required while others are optional. Missing any required function call or mistake the order of some function calls results in a non-executable search, while missing optional function calls lead to an unsuccessful search. We include 120 queries in the test set and provide 11 demonstration examples.

### A.5 GOOGLE SHEETS

This task is to manipulate the real worksheets from the Google Sheets[10], via the gspread library[11]. We include 100 distinct API function calls from the gspread library, but we only create tests for the most common use cases, including updating cell values, sorting, adding or deleting rows and columns, merging cells, filtering, formatting and creating pivot tables. There are 70 test cases and 10 examples in total. We also encourage the model to utilize Pandas DataFrame[12] and gspread-dataframe[13] for advanced manipulations, by explicitly providing 8 additional API functions and certain examples for them. The manipulation is considered as correct only if both the value and the format of each cell match the expectation.

### A.6 VIRTUAL HOME

This task is inherited from the setting of the VirtualHome[14] simulator and asks the LLM to generate sequences of actions for completing household activities. We develop API definitions, demonstration examples, and a test set based on the list of available examples[15] curated in [30]. The API consists of 40 functions, each of which corresponds to a specific action used in the examples. These functions can take up to two arguments, and we collect the list of valid object names for each argument based on all examples. Some examples of the functions include `Sleep()`, `Push(object)`, and `PourInto(object1, object2)`.

The original example list contains 202 household activities, represented by 5088 examples, with each example being a series of actions to complete a specific activity. However, some activities have exactly the same solution as another activity. After deduplication, we are left with 183 unique activities with non-overlapping solutions between any two activities. We randomly select 100 activities to form the test set, while the remaining 83 tasks with their 512 solutions are used as demonstration examples.

When evaluating the LLM's generation for a given task, we consider both executability and correctness. The generation is considered executable if it can be correctly parsed into a series of valid actions, where each action involves only recognizable objects. Regarding correctness, we measure the similarity between the generated program and the ground-truth solution, using the longest common subsequence (LCS) [29] normalized by the maximum length of the two. For tasks with multiple solutions, we consider the highest LCS score from any solution.

---

[10]https://www.google.com/sheets/about/

[11]https://docs.gspread.org/

[12]https://pandas.pydata.org/docs/reference/api/pandas.DataFrame.html

[13]https://gspread-dataframe.readthedocs.io/en/latest/

[14]http://virtual-home.org/

[15]https://github.com/huangwl18/language-planner/blob/main/src/available_examples.json

## A.7    WEBSHOP

This is a multi-step task inherited from Webshop [31], a simulated online shopping environment. The task requires an agent to navigate through a series of webpages to find and purchase a desired product based on a text instruction that outlines the item description.The agent can perform two primary types of actions: `search[text]`, which involves entering a text query, and `click[button]` which involves selecting a button on the page.

We generate demonstration examples based on this file[16], which contains trajectories collected from humans performing the online shopping tasks. We formulate each trajectory into a series of (instruction, webpage description, action) tuples in plain text format. The Long version of the demonstration set consists of 1533 full trajectories, which often exceed the input sequence length limit of the LLM. To address this issue, we provide a Short version of the demonstration examples, by first removing 80% of the non-targeted items from any webpage description, and selecting only the 200 shortest trajectories from the complete set.

For evaluation, we use the predefined simple mode of the WebShop environment[17] and set up the environment with the provided option of using only 1000 random products. We include 100 instructions from sessions with ID numbers 0 to 99 in the test set. We define success as making a purchase which receives a positive reward from the environment within 25 steps.

## A.8    TABLETOP

This task is developed based on the simulated tabletop manipulation domain presented by [50] and outlined in their Appendix K. In this simulation environment, a UR5e robot with a Robotiq 2F85 jaw gripper can perform pick and place actions parameterized by 2D top-down positions. We reuse their API definitions and prompts as demonstration examples. We iterate on the 14 instruction templates used in their evaluation benchmark and create 15 types of tasks that involve manipulating up to 4 colored blocks and 4 colored bowls. For each type of task, we generate 7 valid initial setups of blocks and bowls for the test set, ensuring that no collisions occur during the execution of a valid solution. The success of the LLM's generated program is determined by whether all objects are within a small threshold of their target positions after execution.

# B    COMPREHENSIVE MODEL EVALUATION ON THE TOOLBENCH

In this section, we want to compare the performance of different models on the ToolBench. Specifically, we selected 27 representative LLMs from both closed and open-source community, and evaluate them on the ToolBench in 3-shot scenario.

## B.1    SYSTEM PROMPT

We regularize open-source LLMs to exclusively generate API calls with a system prompt in Figure 5, where the black part is the template shared across all tasks and the red rows are instantiated during inference for a certain goal. Our system prompt first defines a format that combines text sections containing goals, demonstrations, and generations. It then provides explicit guidelines in natural language, instructing the LLMs to generate code exclusively. The system prompt incorporates the goal description and the retrieved API functions directly for each request, reducing the human development effort to a one-time task.

Figure 5: System prompt with guidelines to only generate code in a desired format.

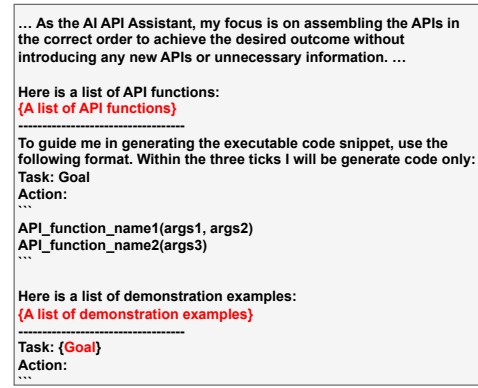

---

Table 7: The achitecture and training data of all the models in our evaluation. The models are grouped by their architecture and training data.

| Model | Architecture | | | Data | | |
|---|---|---|---|---|---|---|
| | Family | Size | Max SS | # Tokens | Pretraining | Finetuning |
| *Closed-source* | | | | | | |
| text-davinci-003 | gpt3 | 175b | 4096 | - | - | - |
| gpt-3.5-turbo | gpt3 | - | 4096 | - | - | - |
| text-curie-001 | gpt3 | 6.7b | 2048 | - | - | - |
| gpt4 | gpt4 | - | 8192 | - | - | - |
| *Open-source* | | | | | | |
| bloomz | bloom | 176b | 2048 | 366B | bloom corpus | xP3 |
| llama-65b | llama | 65b | 2048 | 1.4T | CCNet, C4, | - |
| llama-30b | llama | 30b | 2048 | 1.4T | GitHub, Wikipedia, | - |
| llama-13b | llama | 13b | 2048 | 1.4T | Books, ArXiv, | - |
| llama-13b-alpaca | llama | 13b | 2048 | 1.4T | Stack Exchange | GPT-4 responses, Alpaca |
| starcoderbase | bigcode | 15.5b | 8192 | 1T | The Stack | - |
| starcoder | bigcode | 15.5b | 8192 | 1T | | The Stack (Python) |
| opt-30b | opt | 30b | 2048 | 300B | The Pile, BookCorpus, | - |
| opt-1.3b | opt | 1.3b | 2048 | 300B | CC-Stories, Reddit, | - |
| opt-iml-30b | opt | 30b | 2048 | 300B | CCNewsV2 | OPT-IML Bench |
| opt-iml-1.3b | opt | 1.3b | 2048 | 300B | | |
| gpt-neox-20b | neox | 20b | 2048 | 450B | | - |
| GPT-NeoXT-Chat-Base-20B | neox | 20b | 2048 | 460B | | OpenChatKit IT |
| codegen-16B-nl | neox | 16b | 2048 | 700B | | - |
| codegen-16B-multi | neox | 16b | 2048 | 1T | The Pile | BigQuery |
| codegen-16B-mono | neox | 16b | 2048 | 1T | | BigQuery, BigPython |
| pythia-12b | neox | 12b | 2048 | 300B | | - |
| dolly-v2-12b | neox | 12b | 2048 | 300B | | Dolly IT |
| pythia-6.9b/2.8b1.4b | neox | multi | 2048 | 300B | | - |
| stablelm-base-alpha-7b | neox | 7b | 4096 | 800B | | - |
| stablelm-base-alpha-3b | neox | 3b | 4096 | 800B | The Pile (1.5T) | - |
| stablelm-tuned-alpha-7b | neox | 7b | 4096 | 800B | | Alpaca, GPT4All, |
| stablelm-tuned-alpha-3b | neox | 3b | 4096 | 800B | | Anthropic, Dolly, ShareGPT |

## B.2 MODELS

As listed in Table 7, we select a set of representative LLMs from both closed-source and open-source community.

The closed models are the (Generative Pre-trained Transformer) GPT series from OpenAI, especially the GPT-3[22] and its successors[13]. GPT-3 is a state-of-the-art language model developed by OpenAI, with 175 billion parameters, making it the largest and most powerful language model ever created. It is capable of performing a wide range of natural language processing tasks and has the potential to revolutionize the way we interact with and understand language. Due to the lack of detailed information about its training, we are motivated to study methods to build models achieving similar capabilities, especially using open-source models.

We select the representative and the most advanced open-source models from recent years in our work. They are all decoder-only models, based on transformers[51] architecture. Bloomz[52] is the largest open-source LLM built upon the large-scale multilingual pretrained BLOOM[53]. Bloomz is funtuned on xP3[52], a crosslingual task mixture, for crosslingual generalization to unseen tasks and languages. StarCoder[33] is a family of models developed for purely code generation and synthesis with 8K context length. They exhibit superior performance on common code generation benchmarks. LLaMA[32] is a family of pretrained models, that are performant on quite a few NLP benchamrks. Although they are not as large as Bloomz, they are all trained for almost 4 times longer than Bloom. This is an important reason why they are able to outperform several top peer models on many NLP tasks. Alpaca [54, 55] is fine-tuned LLaMA-13b model on 52K instruction-following data as well responses from GPT-4. OPT-IML[56] is the finetuned version of the original OPT[57], which is the first family of large-scale (176 billion parameters) open-source models that are trained on publicly available datasets. OPT-IML significantly improves the instruction following capability of

OPT by training on a large benchmark of 2000 NLP tasks for Instruction MetaLearning (IML). We only select the publicly accessibile checkpoints from the OPT families in our work.

Another important family of models are all developed from the NeoX toolkit[58] and pretrained using the PILE dataset[59]. GPT-NeoX-20B[60] is only pretrained on the PILE, while GPT-NeoXT-Chat-Base-20B[7] is further finetuned on the OIG-43M[61], a dataset targetting better instruction following capability. CodeGen family[34] is designed for superior capability on code generation, as they are heavily finetuned on large code datasets. Pythia family[62] is a suite of models designed for analyzing LLMs across training and scaling. They are all pretrained on the Pile in the same way, but have different model sizes and intermediate checkpoints released during training. We use those variants in our ablation study. Dolly[63] is finetuned beyond Pythia-12b on a new, high-quality human generated instruction following dataset, crowdsourced among Databricks employees. The StableLM family[64] is pre-trained on an experimental version of the PILE datasets which has 1.5 trillion tokens in total. The models have a sequence length of 4096 to push beyond the context window limitations of the existing open-source language models. The instruction tuned counterpart of each model is also released. By the time we publish this work, only 7b and 3b models are released, while the team behind them is training larger models.

There are other notable models, such as FlanT5[28], the T0 family[23], and the T5 family[65], that have shown promising performance. We do not include all of them in our baseline comparison, as some of their features are not designed for the task at hand. For example, their tokenizers do not distinguish between spaces, tabs and new lines, making it hard for them to generate executable code based on API function calls.

### B.3 EVALUATION

To collect the baseline results, we exploit the naive approach described in section 2 as the action generator. We give each LLM sufficient max tokens to generate on each task and retrieve as many API functions as possible in the prompt. The detailed information is listed in Table 8. We evaluate all the models on a mixture of GPUs and RDUs[66, 67, 68]. In particular, the 176b-parameter bloomz is evaluated on RDU, while all the other models are evaluated on NVIDIA A100 GPUs with 80GB RAM.

For these models, We only conduct the few-shot evaluation described Section 6 because 1) zero-shot results are not representative, as most of them are zero, 2) it is not practical to tune all the models on our training data, and 3) few-shot results can be used as a great proxy of the model performance in all the other settings. For the conversation-oriented models, including gpt-3.5-turbo, chavinlo/gpt4-x-alpaca, GPT-NeoXT-Chat-Base-20B and dolly-v2-12b, we additionally add `<human>:` and `<bot>:` key words in the prompt to better align with their training data format for better performance.

After we get the completion from the LLMs given a prompt, only minimal post-processing steps are applied to the completion: 1) Properly truncate the completion, given the list of task-specific stop sequences and 2) Replace the `{API_KEY}` keywords in the completion with the real API key, so as to execute the code properly. Finally, as shown in Figure 1, to validate the action generated for the single-step tasks, we execute the generated API calls and compare its output against the ground truth; while for the multi-step tasks, the actions are used to interact with the environment directly and only the final status is evaluated. For each task, we report the metrics described in Section 5 for each task. Note that we only evaluate the top 1 generated action with sampling disabled. This is because, in practice, action can only be executed once and there is no chance to reset things and try another action.

### B.4 TOOLBENCH PERFORMANCE OF DIFFERENT MODELS

The performance of different models are summarized in Table 8. Below we show several observations.

**Capability Gap**  Currently, the GPT family of models stands out as the leading players in the field, and there is a significant gap between GPT-4, GPT-3.5 and all the other open-source

Table 8: The performance on ToolBench of different models in 3-shot scenario. The models are group by their architecture and training data.

| Task | Open Weather | The Cat API | Home Search | Trip Booking | Google Sheets | VirtualHome | WebShop Long | WebShop Short | Tabletop |
|---|---|---|---|---|---|---|---|---|---|
| max tokens to generate | 128 | 128 | 128 | 300 | 256 | 128 | 128 | | 256 |
| num API function | all | all | all | all | 10 | 10 | all | | 0 |
| **Closed-source** | | | | | | | | | |
| gpt4 | 93.0 | 96.0 | 97.0 | 96.7 | 62.9 | 23.0 / 23.5 | 0.0 | | 81.0 |
| text-davinci-003 | 99.0 | 98.0 | 97.0 | 89.2 | 62.9 | 31.0 / 25.1 | 0.0 | | 66.7 |
| gpt-3.5-turbo | 90.0 | 92.0 | 80.0 | 85.8 | 51.4 | 20.0 / 18.9 | 0.0 | 1.8 | 33.3 |
| text-curie-001 | 8.0 | 58.0 | 6.0 | 6.7 | 1.4 | 12.0 / 4.1 | 0.0 | 0.0 | 1.0 |
| **Open-source** | | | | | | | | | |
| Llama-2-70b-hf | 90.0 | 84.39 | 83.0 | 71.67 | 58.57 | 35.0 / 24.74 | 1.53 | 30.45 | 45.4 |
| Llama-2-13b-hf | 85.0 | 77.0 | 68.0 | 53.33 | 30.0 | 33.0 / 21.67 | 0.6 | 31.67 | 23.81 |
| Llama-2-7b-hf | 76.0 | 83.0 | 58.0 | 33.33 | 22.86 | 25.0 / 21.49 | 0.0 | 6.92 | 14.39 |
| llama-65b | 90.0 | 80.0 | 84.0 | 65.8 | 32.9 | 32.0 / 20.3 | 0.0 | 41.2 | 30.5 |
| llama-30b | 78.0 | 84.0 | 66.0 | 45.0 | 37.1 | 27.0 / 21.7 | 0.0 | 30.6 | 34.3 |
| llama-13b | 70.0 | 74.0 | 45.0 | 35.8 | 5.7 | 28.0 / 18.9 | 0.0 | 27.6 | 17.1 |
| llama-13b-alpaca | 62.0 | 43.0 | 44.0 | 40.8 | 11.4 | 1.0 / 1.6 | 0.0 | 2.7 | 9.5 |
| starcoder | 91.0 | 84.0 | 82.0 | 51.7 | 48.0 | 23.0 / 19.4 | 2.6 | 0.0 | 21.9 |
| starcoderbase | 90.0 | 86.0 | 79.0 | 63.3 | 42.9 | 24.0 / 16.3 | 5.8 | 23.1 | 17.1 |
| codegen-16B-nl | 51.0 | 75.0 | 37.0 | 21.7 | 7.1 | 43.0 / 18.0 | 0.0 | 0.0 | 16.2 |
| codegen-16B-multi | 56.0 | 75.0 | 47.0 | 7.5 | 21.4 | 31.0 / 14.1 | 0.0 | 0.5 | 8.6 |
| codegen-16B-mono | 63.7 | 72.0 | 52.0 | 28.3 | 31.5 | 28.0 / 15.7 | 1.5 | 6.6 | 15.2 |
| bloomz | 58.0 | 85.0 | 36.0 | 22.5 | 14.3 | 9.0 / 4.9 | 0.0 | 1.0 | 1.0 |
| opt-iml-30b | 44.0 | 48.0 | 5.0 | 3.3 | 2.9 | 13.0 / 8.3 | 0.0 | 0.0 | 1.0 |
| opt-30b | 46.0 | 35.0 | 2.0 | 3.3 | 8.6 | 24.0 / 11.7 | 0.0 | 0.0 | 1.0 |
| opt-iml-1.3b | 20.0 | 28.0 | 0.0 | 0.0 | 4.3 | 13.0 / 3.1 | 0.0 | 0.0 | 1.0 |
| opt-1.3b | 18.0 | 30.0 | 0.0 | 0.0 | 1.4 | 31.0 / 9.7 | 0.0 | 0.0 | 1.0 |
| neox-20b | 55.0 | 69.0 | 27.0 | 10.8 | 18.6 | 28.0 / 15.3 | 0.0 | 8.8 | 6.7 |
| GPT-NeoXT-Chat-Base-20B | 43.0 | 73.0 | 28.0 | 10.8 | 4.3 | 26.0 / 13.1 | 0.0 | 0.7 | 7.6 |
| pythia-12b | 53.0 | 65.0 | 12.0 | 0.8 | 11.4 | 17.0 / 12.1 | 0.0 | 0.0 | 1.9 |
| dolly-v2-12b | 0.0 | 1.0 | 10.0 | 5.0 | 7.1 | 11.0 / 8.9 | 0.0 | 0.0 | 7.6 |
| pythia-12b | 53.0 | 65.0 | 12.0 | 0.8 | 11.4 | 17.0 / 12.1 | 0.0 | 0.0 | 1.9 |
| pythia-6.9b | 41.0 | 72.0 | 8.0 | 7.5 | 4.3 | 29.0 / 14.0 | 0.0 | 0.0 | 8.6 |
| pythia-2.8b | 49.0 | 54.0 | 7.0 | 3.3 | 12.9 | 24.0 / 14.8 | 0.0 | 0.0 | 7.6 |
| pythia-1.4b | 37.0 | 48.0 | 4.0 | 5.0 | 10.0 | 22.0 / 10.7 | 0.0 | 5.2 | 7.6 |
| stablelm-base-alpha-7b | 22.0 | 47.0 | 0.0 | 0.0 | 4.3 | 28.0 / 10.3 | 0.0 | 0.0 | 2.9 |
| stablelm-tuned-alpha-7b | 23.0 | 38.0 | 0.0 | 0.0 | 1.4 | 26.0 / 7.3 | 0.0 | 0.0 | 3.8 |
| stablelm-base-alpha-3b | 6.0 | 28.0 | 0.0 | 0.0 | 1.4 | 29.0 / 5.3 | 0.0 | 0.0 | 1.0 |
| stablelm-tuned-alpha-3b | 14.0 | 31.0 | 0.0 | 0.8 | 0.0 | 8.0 / 5.6 | 0.0 | 0.0 | 1.0 |

Table 9: The statistics of model alignment data

| Task | Open Weather | The Cat API | Home Search | Trip Booking | Google Sheets | VirtualHome | WebShop | Tabletop |
|---|---|---|---|---|---|---|---|---|
| Templates | 90 | 40 | 100 | 30 | 1 | 1 | 2 | 1 |
| Repeat | 20 | 45 | 18 | 60 | 118 | 512 | 900 | 74 |
| Training samples | 1800 | 1800 | 1800 | 1800 | 118 | 512 | 1800 | 74 |
| Complexity score | 1.1 | 1.0 | 6.4 | 10.1 | 12.1 | 12.3 | 0.0 | 4.6 |

models. While open-source models may demonstrate competitiveness on some simpler tasks, they lag far behind on more challenging tasks such as Google Sheets and Tabletop.

**Instruction tuning on conventional NLP tasks doesn't help**  Comparing the models between chavinlo/gpt4-x-alpaca and LLaMA-13b, OPT-IML and OPT, StableLM-tuned and StableLM-base, NeoX-Chat-Base-20b and NeoX, and dolly and pythia, the former model in each pair is intentionally optimized to enhance instruction following capability compared to the latter model. However, no significant accuracy improvement is observed on the ToolBench. Further, the LLaMA family, despite not undergoing any specific instruction tuning during training, still achieves relatively good quality compared to other public models.

**Model size is important**  By comparing the performance of models from GPT faimily, LLaMA family, OPT family, Pythia family and StableLM family, we can clearly see the trend that the larger models tend to perform better on the ToolBench, given the same quantity and quality of their training data.

**Code generation is important**  StarCoder and CodeGen faimily stand out among other models with similar sizes on ToolBench, while StarCoderBase is even on par with the llama-65b model which is more than 4 times larger in size. CodeGen-16B-mono is overall better than its base model CodeGen-16B-nl, which is not specifically tuned for code generation. It is also surprisingly better than CodeGen-16B-multi on almost all the tasks, indicating that it is highly beneficial for action generation if the model is heavily tuned on Python-style code generation.

## C  EXPERIMENT DETAILS

In this section, we extended Section 6 with more details about model training and results.

### C.1  TRAINING DATA

For the OpenWeather, The Cat API, Trip Booking, and Home Search tasks, we generate the training data by converting or expanding the demonstration examples of each task into templates and populating them with various sets of variable values. For the remaining four tasks, we format the training samples directly from the demonstration example set described in section 5. We exclude any test samples from the training data and minimize the overlap of the API function call combinations between any training and test samples. For example, we make sure that the API function combinations used in each test case for the Home Search task are never present in the training data. However, for the OpenWeather task, it was unavoidable to have some overlap because each test case only involved a single function call and the training examples covered all the API functions. The numbers of templates and training samples for each task are summarized in table 9. Example templates and variable values are shown in table 10. The training sets for all tasks, except for the Google Sheets and WebShop task, reduce the complexity score of their respective test sets when compared to the example sets. As expected, the model's accuracy shows improvement after fine-tuning.

### C.2  ALL-SHOT LOSS

To construct the training samples, we concatenate API documents and multiple pairs of goal and API calls as one input sequence to the LLMs. We use an all-shot loss formulation illustrated in Figure 6 which learns to generate the API calls for every goal in a sequence. We use this loss formulation because it empirically delivers better success rate, especially

Table 10: Training template examples of different tools

| | Goal | Action | Variable values |
|---|---|---|---|
| **Open Weather** | What is the present weather situation in {city}? Please respond in {lang} and use {units} units. | curl -X GET 'https://api.openweathermap.org/data/2.5/weather?q={city_formatted}&appid={API_KEY}&lang={lang_abbr}&units={units}' | {city: "Palo Alto", city_formatted: "palo+alto", lang: "English", lang_abbr: "en", units: "imperial"} |
| **The Cat API** | Add the cat photo with id={image_id} to my list of favorites. | curl -X POST 'https://api.thecatapi.com/v1/favourites' –data '{"image_id":"{image_id}"}' | {image_id: "MTUyNTA1OA"} |
| **Home Search** | Looking for homes for sale in {location} with {num_beds} bedrooms and {num_baths} bathrooms, between ${min_price} and ${max_price}. | API.set_location({location})
API.set_buy_or_rent("buy")
API.set_num_beds({num_beds})
API.set_num_baths({num_baths})
API.set_min_price({min_price})
API.set_max_price({max_price})
API.search() | {location: "Palo Alto", num_beds: 4, num_baths: 5, min_price: 7000000, max_price: 8000000} |
| **Trip Booking** | Search for {means_of_transportation} tickets for {num_adults} adults from {location_from} to {location_to}, on {departure_date}. | API.select_booking_type("trip tickets")
API.select_transportation({means_of_transportation})
API.set_num_adults({num_adults})
API.set_origin(Loc({location_from}))
API.set_destination(Loc({location_to}))
date = Date({departure_date})
API.set_departure_date(date)
API.search() | {means_of_transportation: "flight", max_price_ticket: 150, num_adults: 2, location_from: "San Francisco", location_to: "Los Angeles", departure_date: "2023-08-15"} |
| **Google Sheet** | {task} | {action} | {task: "
\| Product \| Cost \| Price \|
\| beef \| 1 \| 3 \|
\| pork \| 5 \| 4 \|
\| chicken \| 10 \| 11 \|
\| lamb \| 3 \| 15 \|
\| duck \| 12 \| 2 \|
\| fish \| 2 \| 100 \|

Task:
Sum B1:B4 and write the result below B4
Action:",
action: "
worksheet.update('B5', '=SUM(B1:B4)', raw=False)"} |
| **VirtualHome** | {task} | {action} | {task: "
Task: Read book
Action:",
action: "
Agent.Find(novel)
Agent.Grab(novel)
Agent.Find(chair)
Agent.SitOn(chair)
Agent.Read(novel)"} |
| WebShop | {task} | {action} | {task: "Instruction: i'm looking to buy a high resolution marine animal themed backdrop. the size should be 12x10ft, and price lower than 100.00 dollars
[button] Search [button_]
Action:",
action: "
search[12x10ft high resolution marine animal backdrop]"} |
| **Tabletop** | {task} | {action} | {task: "objects = ['yellow block', 'green block', 'yellow bowl', 'blue block', 'blue bowl', 'green bowl']
# move the green block to the top right corner.",
action: "
corner_pos = parse_position('top right corner')
put_first_on_second('green block', corner_pos)"} |

Figure 6: We use all-shot loss for model alignment. We concatenate several examples into a single training sample and backpropagate through the loss on the blue actions in every example. There is no separator token between examples.

Table 11: The detailed performance on the ToolBench of models with different techniques applied. Mean(standard deviation) values are provided for each task. There exists some inevitable randomness, but it won't cange the results by too much.

| Task | Open Weather | The Cat API | Home Search | Trip Booking | Google Sheets | VirtualHome | WebShop Long | WebShop Short | Tabletop |
|---|---|---|---|---|---|---|---|---|---|
| **Zero-shot Baseline** | | | | | | | | | |
| gpt4 | 81.3(1.7) | 97.4(0.3) | 76.6(1.1) | 91.5(0.5) | 5.7(0.0) | 40.8(0.6) / 8.0(0.2) | 0.0(0.0) | - | - |
| llama-30b | 39.0(0.0) | 49.0(0.0) | 0.0(0.0) | 0.0(0.0) | 0.0(0.0) | 78.0(0.0) / 0.3(0.0) | 0.0(0.0) | - | - |
| starcoder | 32.0(0.0) | 71.0(0.0) | 7.0(0.0) | 13.3(0.0) | 5.9(1.1) | 22.0(0.0) / 3.7(0.0) | 0.0(0.0) | - | - |
| codegen-16B-mono | 7.0(0.0) | 78.0(0.0) | 0.0(0.0) | 0.0(0.0) | 1.4(0.0) | 4.0(0.0) / 1.0(0.0) | 0.0(0.0) | - | - |
| **Sys. Prompt** | | | | | | | | | |
| gpt4 | 78.4(0.3) | 94.2(0.8) | 72.7(2.0) | 89.6(0.9) | 28.6(0.0) | 42.8(0.6) / 8.6(0.1) | 0.0(0.0) | - | - |
| llama-30b | 50.0(0.0) | 88.0(0.0) | 0.0(0.0) | 0.0(0.0) | 11.4(0.0) | 24.0(0.0) / 2.5(0.0) | 0.0(0.0) | - | - |
| starcoder | 71.0(0.0) | 91.0(0.0) | 2.0(0.0) | 7.5(0.0) | 15.9(0.2) | 26.0(0.0) / 4.9(0.0) | 0.0(0.0) | - | - |
| codegen-16B-mono | 32.0(0.0) | 69.0(0.0) | 0.0(0.0) | 0.0(0.0) | 7.1(0.0) | 5.0(0.0) / 1.6(0.0) | 0.0(0.0) | - | - |
| **3-shot** | | | | | | | | | |
| gpt4 | 93.0(0.0) | 96.0(0.0) | 97.0(0.0) | 96.7(0.0) | 62.9(0.0) | 23.0(0.0) / 23.5(0.0) | 0.0(0.0) | 0.0(0.0) | 81.0(0.0) |
| llama-30b | 78.0(0.0) | 84.0(0.0) | 66.0(0.0) | 45.0(0.0) | 37.1(0.0) | 27.0(0.0) / 21.7(0.0) | 0.0(0.0) | 30.6(0.0) | 34.3(0.0) |
| starcoder | 91.0(0.0) | 84.0(0.0) | 82.0(0.0) | 51.7(0.0) | 48.0(1.1) | 23.0(0.0) / 19.4(0.0) | 2.6(0.0) | 0.0(0.0) | 21.9(0.0) |
| codegen-16B-mono | 63.7(0.5) | 72.0(0.0) | 52.0(0.0) | 28.3(0.0) | 31.5(0.5) | 28.0(0.0) / 15.7(0.0) | 1.5(0.0) | 6.6(0.0) | 15.2(0.0) |
| **Alignment** | | | | | | | | | |
| llama-30b | 100.0(0.0) | 94.0(0.0) | 85.0(0.0) | 87.5(0.0) | 4.3(0.0) | 14.0(0.0) / 10.6(0.0) | 20.8(0.0) | - | - |
| starcoder | 95.0(0.0) | 98.0(0.0) | 78.0(0.0) | 85.0(0.0) | 10.0(0.0) | 28.0(0.0) / 13.4(0.0) | 0.0(0.0) | - | - |
| codegen-16B-mono | 99.0(0.0) | 95.8(0.6) | 78.0(0.0) | 73.3(0.0) | 10.0(0.0) | 10.0(0.0) / 11.5(0.0) | 30.3(0.0) | - | - |
| **Sys. Prompt + 3-shot** | | | | | | | | | - |
| gpt4 | 99.0(0.0) | 98.0(0.0) | 98.0(0.0) | 99.2(0.0) | 68.6(0.0) | 29.0(0.0) / 21.7(0.0) | 0.0(0.0) | 0.0(0.0) | 83.8(0.0) |
| llama-30b | 66.0(0.0) | 82.0(0.0) | 63.0(0.0) | 45.8(0.0) | 27.1(0.0) | 34.0(0.0) / 20.5(0.0) | 0.0(0.0) | 0.0(0.0) | 34.6(0.2) |
| starcoder | 92.0(0.0) | 91.0(0.0) | 73.0(0.0) | 54.2(0.0) | 50.0(0.2) | 28.0(0.0) / 15.0(0.0) | 0.0(0.0) | 0.0(0.0) | 23.4(0.3) |
| codegen-16B-mono | 64.2(0.3) | 70.0(0.0) | 45.0(0.0) | 22.5(0.0) | 28.6(0.9) | 27.0(0.0) / 15.7(0.0) | 0.0(0.0) | 0.0(0.0) | 14.6(0.2) |
| **Sys. Prompt + Alignment** | | | | | | | | | |
| llama-30b | 100.0(0.0) | 94.0(0.0) | 79.0(0.0) | 80.8(0.0) | 5.7(0.0) | 10.0(0.0) / 10.3(0.0) | 0.6(0.0) | - | - |
| starcoder | 98.7(0.2) | 97.0(0.0) | 79.0(0.0) | 84.2(0.0) | 10.0(0.0) | 18.0(0.0) / 10.3(0.0) | 0.0(0.0) | - | - |
| codegen-16B-mono | 99.0(0.0) | 96.0(0.0) | 77.0(0.0) | 75.8(0.0) | 8.6(0.0) | 7.0(0.0) / 10.0(0.0) | 25.7(0.0) | - | - |
| **3-shot + Alignment** | | | | | | | | | |
| llama-30b | 100.0(0.0) | 94.0(0.0) | 88.0(0.0) | 89.2(0.0) | 4.3(0.0) | 20.0(0.0) / 26.3(0.0) | 19.5(0.0) | 15.1(0.0) | 6.9(0.2) |
| starcoder | 100.0(0.0) | 96.0(0.0) | 91.0(0.0) | 84.2(0.0) | 15.7(0.0) | 48.0(0.0) / 21.3(0.0) | 0.0(0.0) | 0.0(0.0) | 13.9(0.3) |
| codegen-16B-mono | 99.0(0.0) | 97.9(0.2) | 80.0(0.0) | 77.5(0.0) | 16.4(0.9) | 38.0(0.0) / 18.6(0.0) | 6.5(0.0) | 17.5(0.0) | 16.2(0.0) |
| **Prompt + 3-shot + Alignment** | | | | | | | | | |
| llama-30b | 100.0(0.0) | 94.0(0.0) | 87.0(0.0) | 85.8(0.0) | 2.9(0.0) | 16.0(0.0) / 24.3(0.0) | 0.0(0.0) | 0.0(0.0) | 7.5(0.1) |
| starcoder | 99.0(0.0) | 97.0(0.0) | 83.0(0.0) | 80.8(0.0) | 21.2(0.3) | 31.0(0.0) / 18.4(0.0) | 0.0(0.0) | 0.0(0.0) | 13.9(0.3) |
| codegen-16B-mono | 97.7(0.2) | 99.0(0.0) | 82.0(0.0) | 77.5(0.0) | 19.8(0.3) | 29.0(0.0) / 17.2(0.0) | 0.0(0.0) | 3.5(0.0) | 16.2(0.0) |

when using in-context demonstrations, than the conventional loss which only backpropagates the loss associated with the API calls for the last goal.

### C.3 Training details

We finetune each model on the same dataset created with the method described in Section C.1 for 8 epochs. We use a max sequence length of 2048 without packing and mix the data from all the tasks into a single dataset with random shuffling. In each sample, all the goal-action pairs are from the same task. We report the validation accuracy on the best checkpoint. We use a batch size of 16 and a constant learning rate of $1e-5$ for each model and train on an internal cluster of 4 A100 GPU's, each with 80GB RAM.

### C.4 Extended results for Section 6

We list out the detailed results of Section 6 in Table 11, where we report the model performance on all the possible combinations of the three proposed techniques. The main observations are all covered in Section 6. We run each job 3 times, and report the mean and standard deviation of the main metrics. Their are some inevitable randomness happens in API or example retrieval, public API services and the environment provided in Webshop

and Tabletop. Even though randoness exists, we observe that they barely change the final results. Thus, we only report the mean value everywhere else in the paper.

# D    API Selection Complexity Score

## D.1    Complexity score

This section introduces a complexity score system designed to measure the intrinsic complexity and difficulty of the tasks from *ToolBench*. The complexity score system aims to provide a quantitative measure of the intrinsic complexity of the tests given the examples by calculating the probability of the tests being derived or converted from the examples; and the derivation or conversion is performed in a random system with all possible outcomes equally likely. This score serves to assess the inherent level of difficulty involved in transitioning from one scenario to another, thereby assisting researchers and developers in benchmark evaluation and analysis.

### D.1.1    The likelihood of a test being derived from an example

In the complexity score system proposed herein, the calculation of the complexity score involves assessing the probability or likelihood of the tests being derived from an example in the particular task. Given a demonstration example $e$ and a set of API functions $\mathcal{D}$, the derivation of a particular test sample $t$ involves two major steps: 1) remove all the unused API calls while keeping all the necessary ones and 2) insert the new API calls that $e$ does not cover. Given a random system, where all possible outcomes are equally likely, we suppose the deletion possibility of each API call from $e$ is 50%, while the insertion possibilities of the correct API call is $1/|\mathcal{D}|$, where $|\mathcal{D}|$ is the total number of API functions of the given task. If $t$ or $e$ contains multiple calls to the same API function, we consider them as different API calls, because they are usually not interchangeable. Based on these assumptions, the likelihood of generating a test sample $t$ is calculated using Equation (1).

$$
p(t \mid e, \mathcal{D}) = \left(\frac{1}{2}\right)^{|e|} \left(\frac{1}{|\mathcal{D}|}\right)^{|t \backslash e|}
\tag{1}
$$

where $|e|$ represents the number of API calls in the example $e$, and $|t \backslash e|$ is the number of uncovered API calls in the test sample. Suppose we have a task that has 10 API functions in total $\{a_i\}_1^{10}$, and the demonstration example covers $\{a_1, a_2, a_3, a_4\}$, but the test sample requires $\{a_1, a_2, a_6, a_4, a_5\}$. In the first step, the probability of successfully dropping $a_3$ while keeping the rest ones in $e$ is $\left(\frac{1}{2}\right)^4$. Then, the probability of correctly adding in the uncovered ones, $a_5$ and $a_6$, is $\left(\frac{1}{10}\right)^2$. Note that we do not take the order of API calls into consideration for the purpose of being simple without losing generosity.

### D.1.2    The distance between a test and example pair

We first define the distance $d$ between one particular test and example pair by take the logarithm of the reciprocal of Equation (1) as:

$$
d(t, e) = \log\left[\frac{1}{p(t \mid e, \mathcal{D})}\right]
\tag{2}
$$

The use of the reciprocal in the expression aligns the complexity score with the definition of complexity, where a higher score indicates a greater level of complexity. Additionally, applying the logarithm to the reciprocal value aids in addressing the magnitude gap. The logarithm function compresses the range of values, reducing the impact of extreme values and creating a more manageable scale. This normalization ensures that the complexity score is not disproportionately influenced by outliers or extreme values, providing a more balanced representation of complexity across the range of input values. By combining the reciprocal and logarithm, the expression effectively balances the score by aligning it with the definition of complexity and mitigating the impact of magnitude differences in the input values.

### D.1.3 COMPLEXITY SCORE OF A TASK

Based on the complexity score of generating a test from an example, we can construct the complexity score $S$ of a given task. The score $S = f(\mathcal{T}, \mathcal{X}, \mathcal{D})$ is a function of the test samples $\mathcal{T}$, the demonstration examples $\mathcal{X}$ and the API functions $\mathcal{D}$ of each task.

$$
\begin{aligned}
S(\mathcal{T}, \mathcal{X}, \mathcal{D}) &= \mathbb{E}_{t \in \mathcal{T}} \min_{e \in \mathcal{X}} d(t, e) \\
&= \mathbb{E}_{t \in \mathcal{T}} \min_{e \in \mathcal{X}} \log \left[ \frac{1}{p(t \mid e, \mathcal{D})} \right] \\
&= -\mathbb{E}_{t \in \mathcal{T}} \max_{e \in \mathcal{X}} \log \left[ \left( \frac{1}{2} \right)^{|e|} \left( \frac{1}{|\mathcal{D}|} \right)^{|t \backslash e|} \right]
\end{aligned}
\tag{3}
$$

This score ranges from zero to infinity. The larger the score is, the more challenging a task is in terms of API selection. We calculate this score for both the original ToolBench (Table 3) and the training data we created for alignment Table 9. They share the same $\mathcal{D}$ and $\mathcal{T}$, but have a different $\mathcal{X}$, so that their API selection complexities are different for each task.

### D.2 COMPLEXITY SCORE ON THE TOOLBENCH

In this section we demonstrate how the complexity score behaves on the ToolBench.

### D.2.1 COMPUTATION DETAILS

For the Trip Booking, Home Search, Virtual Home, and Google Sheets tasks, the set of API functions $\mathcal{D}$ is the same as described in appendix A. For the single-step, single-API-call tasks, Open Weather and The Cat API, each valid URL with parameters is treated as a unique API option in set $\mathcal{D}$. In total, Open Weather has 37 API options, while The Cat API has 52 API options. In the case of the Tabletop task, since there are no predefined correct answers for the test cases, we divide the three set of "Tabletop Manipulation" examples[18] into 65 single-step samples. Note that for the WebShop task, since there are only two API functions always covered by the example set, the complexity score is 0 by definition.

### D.2.2 REVERSED-F1 SCORE

For comparison purpose, we also consider the simple Reversed-F1 (r-F1) distance $d_{r-F1}$, derived from the conventional F1 score[69], between one particular test and example pair as

$$
d_{r-F1}(t, e) = (1 - F1(t, e)) * 100
\tag{4}
$$

We multiply 100 to the score to align with the range of the complexity score defined above. Follow the same definition proposed in appendix D.1.3, we can construct the r-F1 score $S_{r-F1}$ of a given task as:

$$
\begin{aligned}
S_{r-F1}(\mathcal{T}, \mathcal{X}) &= \mathbb{E}_{t \in \mathcal{T}} \min_{e \in \mathcal{X}} d_{r-F1}(t, e) \\
&= \mathbb{E}_{t \in \mathcal{T}} \min_{e \in \mathcal{X}} [(1 - F1(t, e)) * 100]
\end{aligned}
\tag{5}
$$

### D.2.3 MEASUREMENTS

In this section, Spearman's Correlation Coefficient (SCC) [70] is employed to assess the effectiveness of the proposed complexity score. The evaluation involves the analysis of five different tasks using three models: GPT-4, LLaMA-30b, CodeGen-16b, and StarCoder. We only include the five tasks without

Table 12: Spearman's Correlation Coefficients

|  | GPT-4 | LLaMA | CodeGen | StarCoder |
|---|---|---|---|---|
| Complexity | 0.2 | 1.0 | 1.0 | 0.7 |
| r-F1 | -0.3 | 0.7 | 0.7 | 0.3 |

---

[18]https://code-as-policies.github.io/

Figure 7: Spearman's correlation coefficient(SCC) is computed separately for two comparisons: (1) complexity score and error rate, and (2) reversed F1 score and error rate on five tasks: (1) Open Weather, (2) The Cat API, (3) Home Search, (4) Trip Booking, and (5) Virtual Home.

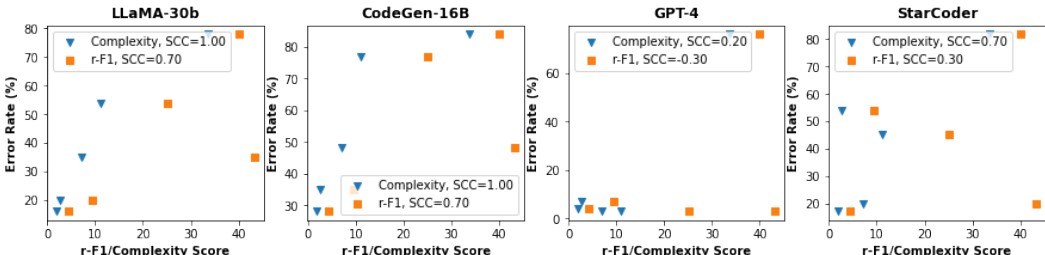

advanced reasoning from table 3, as the advanced reasoning breaks the correlation between the API selection difficulty and the final model performance. The complexity score and the r-F1 score are calculated for each task. SCC is then computed separately for two comparisons: (1) complexity score and error rate, and (2) reversed F1 score and error rate, for all five tasks. The results are illustrated in fig. 7 and table 12.

The findings of the study reveal near-perfect Spearman's correlation coefficient (SCC) between the complexity score and the error rate for the LLaMA-30b, CodeGen-16b and StarCoder models. This strong correlation indicates that the proposed complexity score system accurately captures the intrinsic difficulty of these tasks.

For more powerful models like GPT4, which exhibit near-perfect accuracy (above 93%) for low-complexity tasks (complexity $< 12$) such as Open Weather, The Cat API, Home Search, and Trip Booking, the SCC becomes relatively sensitive to any randomness or turbulence during the experiments. Consequently, the complexity score system shows a non-perfect SCC of 0.2 in this case.

Despite the sensitivity of the SCC in the GPT4 experiments, the complexity score remains a superior indicator of task difficulty compared to the r-F1 score. It effectively captures the inherent difficulty of each task and provides valuable insights into task complexity. Overall, complexity score is more effective at capturing the inherent difficulty of each task, thus providing valuable insights into task complexity.

The obtained results provide empirical evidence supporting the validity and reliability of the proposed complexity score system. The high SCC values signify a consistent relationship between the complexity score and the error rate across different models and tasks. This correlation strengthens the argument that the complexity score accurately captures the complexity and difficulty of the benchmarks, enabling researchers and developers to assess and compare the inherent challenges associated with different tasks.