# OpenReview forum: "On the Tool Manipulation Capability of Open-sourced Large Language Models"
_ICLR.cc/2024/Conference — Submitted to ICLR 2024_

### Official Review · Reviewer_Qyny · 2023-10-24

**Soundness:** 3 good
**Presentation:** 3 good
**Contribution:** 2 fair
**Rating:** 6
**Confidence:** 4

**Summary:**

This paper aims to enhance open-source Large Language Models (LLMs) for tool manipulation. The study addresses the limitations of closed APIs in terms of security and robustness by proposing techniques, such as model alignment with programmatic data generation, system prompts, and in-context demonstration retrievers. Through evaluation on ToolBench, a diverse benchmark of tools, the authors demonstrate that their approach can significantly improve open-source LLMs, achieving up to a 94% success rate and competitive performance with OpenAI GPT-4 in four out of eight tasks.

**Strengths:**

1. Focus on a timely problem for evaluating the LLM's ability of tool usage.

2. The introduction of ToolBench, a benchmark consisting of diverse software tools for real-world tasks, strengthens the paper's contributions.

3. Diagnosis of existing tool-use LLMs to highlight their strengths, weaknesses, and potential improvements.

**Weaknesses:**

1. Missing comparison with closely related works. I would like to inquire about the absence of a comparative analysis with Toolbench, ToolQA and Gorilla (see [1,2,3] for references). An exploration of potential distinctions between the two approaches would provide valuable insights.

2. The retrieval quality in this study appears to be not very good. In practice, I have observed that the retrieved demonstrations often lack meaningful relevance. Such limitations can significantly impact overall system performance.

3. Lacking substantial enhancements tailored specifically for open-source large language models. Have the authors considered the exploration of other decoding strategies or innovative approaches to improve the overall system performance and capabilities?

4. The template of the paper seems to be incorrect for ICLR. Please fix it in the camera-ready version.

5. It is **not appropriate** to include the conference name 'neurips2023' in your GitHub repo. I believe this is the submission site for ICLR 2024, not NeurIPS 2023. Also, it is recommended to use the anonymous github for submitting code.


[1] Qin, Yujia, et al. "Toolllm: Facilitating large language models to master 16000+ real-world apis." arXiv preprint arXiv:2307.16789 (2023).

[2] Zhuang, Yuchen, et al. "ToolQA: A Dataset for LLM Question Answering with External Tools." arXiv preprint arXiv:2306.13304 (2023).

[3] Patil, Shishir G., et al. "Gorilla: Large language model connected with massive apis." arXiv preprint arXiv:2305.15334 (2023).

**Questions:**

See above.

---

> ### Author Response · Authors · 2023-11-15
>
> Thanks for recognizing the value of our work as well as your constructive feedback!
>
> **Regarding the strengths and motivation**
>
> Rather than completely solving the tool using issues for LLMs by proposing new algorithms or models, our main motivation of this paper is more like opening the door for the community to explore and improve models/algorithms on tool using. As you mentioned in the strengths section, we sincerely hope our work can be published at ICLR so that the researchers and practitioners can 1) see and use our *ToolBench* to guide and evaluate their further research, 2) take the techniques we revisited as teasers to encourage further innovation that can fundamentally improve model tool using capability, and 3) exploit those techniques to enhance model performance on their own tools and tasks in practice.
>
> **Regarding the weaknesses and your questions**
>
> 1. **Comparison with other similar works** - Thanks for pointing them out. We will definitely include those concurrent work in the literature review in the following version of the paper. Although those works also study different ways of tools and API use, the tasks they are focusing on are very similar to certain subsets from our ToolBench, since the principal of the *Toolbench* design is to cover tasks with different levels and types of difficulty. Nevertheless, we are more than happy to extend our *Toolbench* to include their tasks if this can further benefit the community.
>
> - The Toolbench from ToolLLaMA paper includes a large collection of APIs, but all of them are only REST APIs, which are pretty similar to the OpenWeather and the CAT API from our ToolBench collection. In order to generalize the tool using capability of LLMs, a good benchmark should include different types of APIs - REST API, Python API, human text command API, etc. This is also one of the main focus of our ToolBench design.
>
> - Gorilla’s APIBench mainly focuses on loading models from 3 major model hubs. Their tasks can also be categorized into the ToolBench evaluation framework. Those tasks have very limited number of API functions in total, and the major difficult lies in populating the proper arguments (e.g. model names). This is also very similar to our WebShop task, where each function call is simply to click a certain button on the webpage, but the difficulty lies in for example selecting the correct product among tens of others to click.
>
> - ToolQA is designed for a different purpose. It focuses only on question answering scenario, and has an additional layer checking whether or not tool uses needs to be invoked and what tool to use. While the other works, including ours, only focusing on evaluating the model capability and accuracy in using certain tools.
>
> 2. **Demonstration examples concerns**
>
> - In practice, for each given tool or a set of API functions, we found it is usually easy to collect a bunch of demonstration examples for most of the typical uses cases. However, the demonstration example may not 100% cover all the potential use cases that need new combinations of API functions to achieve certain goals. To validate this, we conduct experiments in section 4.2, where we made sure that the test cases all require completely different API function combinations than the ones in the 10 demonstration examples. As shown in Figure 4, the models are still able to reach good accuracy with only several examples retrieved, indicating that the models can indeed generalize to unseen API combinations from limited retrieved examples.
>
> - We agree that improving the retriever capability is definitely a promising research direction. But since we mainly revisited those three techniques in our work as proof-of-concepts to show that they are effective to improve model tool using capabilities with limited human supervision, we decided to leave retriever improving as future work.
>
> 3. **Lacking substantial enhancements for the base models** - This is another aspect we leave for future work, as it is outside the scope of this paper. However, we benchmarked quite some open-source models in Appendix section B and Table 8. We have several interesting findings and we found that model size and coding capability especially contribute a lot to the model tool using capabilities. This trend is also confirmed with the newly related LLaMA2 series models and CodeLLaMA models. Thus, we think the *ToolBench* dataset is valuable to the community and can help guide the model development in the long run.
>
> 4. **Paper formatting** - Thanks for the suggestions! We will double check to make sure the format is correct in the next version of the paper. We will also stick with anonymous github link in the future submissions.

---

> > ### Comment · Reviewer_Qyny · 2023-11-15
> >
> > Thanks for your response!

---

### Official Review · Reviewer_Y6vx · 2023-10-25

**Soundness:** 2 fair
**Presentation:** 2 fair
**Contribution:** 2 fair
**Rating:** 5
**Confidence:** 5

**Summary:**

Tool learning is a rising topic with the emergence of powerful closed LLMs (e.g., ChatGPT). However, there still remains a large gap between closed LLMs and open-source LLMs. To reduce the gap, this paper explores how to enhance the capability of open-source LLMs in tool utilization. More specifically, authors argue three challenges for improving open-source LLMs, including usage examples, in-context demonstration, and generation style regulation. To address these, this paper attempts to augment the practical amount of supervision and enhance LLMs by using model alignment, in-context demonstration retriever and system prompt. Moreover, this paper introduces a ToolBench benchmark to address this problem. Experimental results demonstrate that the proposed method can improve the capability of open-source LLMs in tool utilization.

**Strengths:**

1. This paper analyzes the mistakes of current LLMs in API usage.
2. This paper introduces a benchmark, called ToolBench, to investigate the potential of open-source LLMs in tool use. Moreover, authors also introduce API complexity to examine the challenge of the proposed ToolBench.
3. Experimental results demonstrate the effectiveness of the proposed method.

**Weaknesses:**

1. This paper involves human-curated templates to build model alignment. However, such a stage may raise some problems:

+ The curated examples mainly cover 6 - 8 scenarios. So what happens if we extend to another different scenario? Does it require us to produce more curated data? The generalization of these curated examples needs to be verified.
+ Since the curated data are involved, are there any experiments to validate the connection between the quality of curated data and final results (e.g., model alignment)? We need to confirm how curated data affect model performance to what extent. If the value of the proposed method could be reduced if it heavily relied on human-curated data,

2. This paper applies a demonstration retrieval to obtain the most related demonstrations. However, just as aforementioned, such a stage requires us to have some demonstration in before. So what happens if we do not have relevant demonstrations for a request?

**Questions:**

1. Recently, there also have some benchmarks like another ToolBench [1]. Can authors compare or describe the differences among these works?
2. Improving model performance by fine-tuning LMs is not surprising. It is necessary to validate the generalization of tuning LLMs. For example, tuning LLMs on a subset of the dataset (e.g., Open weather) and then evaluating it on the remaining parts (e.g., Virtual Home).

One minor question about the presentation:
1. Can authors check the latex format of the submitted version? It seems that the font style is different from the standard version of the ICLR submission.

[1] ToolLLM: Facilitating Large Language Models to Master 16000+ Real-world APIs.

---

> ### Author Response · Authors · 2023-11-15
>
> Thanks for recognizing the value of our work as well as your constructive feedback!
>
> **Regarding the strengths and motivation**
>
> Rather than completely solving the tool using issues for LLMs by proposing new algorithms or models, our main motivation of this paper is more like opening the door for the community to explore and improve models/algorithms on tool using. As you mentioned in the strengths section, we sincerely hope our work can be published at ICLR so that the researchers and practitioners can 1) see and use our *ToolBench* to guide and evaluate their further research, 2) take the techniques we revisited as teasers to encourage further innovation that can fundamentally improve model tool using capability, and 3) exploit those techniques to enhance model performance on their own tools and tasks in practice.
>
> **Regarding the weaknesses and your questions**
>
> 1. **Finetuning Transferability** - We agree that this is a topic we didn’t discuss in our paper. We include the finetuning technique in our work only as a proof-of-concept to show that it is also effective to improve model tool using capabilities with limited human supervision.  We wanted to keep involving as minimum human supervision as possible in our work. With the training data details in appendix section C.1, the dataset is just prepared by 1 author of us in just one day using the programmatic way described in section 4.1. Since making a perfect model is not the main goal of this paper, we leave the finetuning scaling law study as well as the transferrablity across tasks as future work, where we would really love to work with the community together to explore more robust algorithms and more efficient data collecting methods to fundamentally improve model performance on the ToolBench.
>
> 2. **Demonstration examples concerns**
>
> - In practice, for each given tool or a set of API functions, we found it is usually easy to collect a bunch of demonstration examples for most of the typical uses cases. The tool developers who compile the API documentation usually also include demonstration examples to make the tool more friendly to the future users.
>
> - However, the demonstration example may not 100% cover all the potential use cases that need new combinations of API functions to achieve certain goals. To validate this, we conduct experiments in section 4.2, where we made sure that the test cases all require completely different API function combinations to achieve each goal than the ones in the 10 demonstration examples. As shown in Figure 4, the models are still able to reach good accuracy with several examples retrieved, indicating that the models can indeed generalize to unseen API conbinations from limited retrieved examples.
>
> 3. **Comparison with other similar works** - Thanks for pointing it out. We will definitely include all the concurrent work, including ToolLLM, in the literature review in the following version of the paper. Although those works also study different ways of tools and API use, the tasks they are focusing on are very similar to certain subsets from our *ToolBench*, since the principal of the *Toolbench* design is to cover tasks with different levels and types of difficulties. Nevertheless, we are more than happy to extend our *Toolbench* to include their tasks if this can further benefit the community.
>
> - The Toolbench from ToolLLaMA paper includes a large collection of APIs, but all of them are only REST APIs, which are pretty similar to the OpenWeather and the CAT API from our ToolBench collection. In order to generalize the tool using capability of LLMs, a good benchmark should include different types of APIs - REST API, Python API, human text command API, etc. This is also one of the main focus of our ToolBench design.
>
> 4. **Latex format** - Thanks for capturing this. We will double check to make sure the format is correct in the next version of the paper.

---

> ### Author Response · Authors · 2023-11-22
>
> Dear Reviewer Y6vx,
>
> We sincerely appreciate your valuable and constructive feedback. We would like to kindly remind you that the author/reviewer discussion phase concludes on November 22nd. We hope our responses have addressed your concerns and improved the paper’s quality. If you have any further suggestions or comments, please feel free to share them. We are looking forward to a constructive discussion during the rebuttal phase.
>
> Best Regards,
>
> The Authors

---

### Official Review · Reviewer_3VL8 · 2023-10-30

**Soundness:** 3 good
**Presentation:** 3 good
**Contribution:** 2 fair
**Rating:** 5
**Confidence:** 4

**Summary:**

The paper analyzed the tool manipulation capability of open-source LLMs and pointed out some common failure types: 1) open-source LLMs struggle with accurately recognizing API names, 2) in the absence of few-shot demonstrations, open-source LLM often incorrectly populate API arguments, 3) open-source LLMs usually produce natural language descriptions rather than executable code. Based on these failure types, the author investigated the benefit of three techniques: 1) instruction-tuning LLM with samples about API calling, 2) retrieve demonstrations for in-context learning, and 3) adding a system prompt tailored to guide the model's outputs. To evaluate these techniques, the author proposed a new benchmark suite called ToolBench that covers eight different API usage scenarios. According to the experiments, the author found that the fine-tuning step boosts the tool-use capability of open-source LLM significantly, while system prompt and in-context learning robustify the LLMs for further improvement.

**Strengths:**

The paper is nicely written and has good value for practitioners. The three techniques the paper studied are reasonable and can effectively boost the LLM's capability in using tools, as suggested in Table 5. The author also conducted ablation study on each technique's relative contribution (Table 6).

**Weaknesses:**

The novelty is limited because prior works (such as [NeurIPS2023] GPT4Tool) have already shown that instruction-tuning can boost the tool-use capability of open-source LLMs. Other techniques studied by the paper, such as in-context retriever and system message, are also standard prompting methods. The author has not compared ToolBench with other similar benchmarks like the GPT4Tools dataset and the APIBench proposed in Gorilla (https://arxiv.org/abs/2305.15334).

**Questions:**

1. Coding models seem to perform better at API tasks. However, the author has not compared with CodeLLaMA (https://huggingface.co/codellama).
2. The author gave an example about advanced reasoning in Table-4. In this example, the model needs to understand that in a Google Sheet, the cell that contains the beef's price is "C2". However, what if the column that contains Price is column "D" rather than column "C"? Did you gave additional context to the LLM?

---

> ### Author Response · Authors · 2023-11-15
>
> Thanks for recognizing the value of our work as well as your constructive feedback!
>
> **Regarding the novelty and motivation**
>
> Rather than completely solving the tool using issues for LLMs by proposing new algorithms or models, our main motivation of this paper is more like opening the door for the community to explore and improve models/algorithms on tool using. As you mentioned in the summary and strengths sections, we sincerely hope our work can be published at ICLR so that the researchers and practitioners can 1) see and use our *ToolBench* to guide and evaluate their further research, 2) take the techniques we revisited as teasers to encourage further innovation that can fundamentally improve model tool using capability, and 3) exploit those techniques to enhance model performance on their own tools and tasks in practice.
>
> **Regarding the weaknesses and your questions**
>
> 1. **Comparison with other similar works** - Thanks for pointing them out. We will definitely include those concurrent works in the literature review in the following version of the paper. Although those works also study different ways of tools and API use, the tasks they are focusing on are very similar to certain subsets from our *ToolBench*, since the principal of our *Toolbench* design is to cover tasks with different levels and types of difficulties. Nevertheless, we are more than happy to extend our *Toolbench* to include their tasks if this can further benefit the community.
>
> - GPT4Tools focuses on getting models to operate on a set of image manipulation tools, where each tool is a single function call. To accomplish each task, the model needs to properly finish a series of function calls in order. This is pretty similar to the Home Search, Trip Booking, Google Sheet and Virtualhome from our *ToolBench*.
>
> - Gorilla’s APIBench mainly focuses on loading models from 3 major model hubs. Their tasks can also be categorized into the ToolBench evaluation framework. Those tasks have very limited number of API functions in total, and the major difficult lies in populating the proper arguments (e.g. model names). This is also very similar to our WebShop task, where each function call is simply to click a certain button on the webpage, but the difficulty lies in for example selecting the correct product among tens of others to click.
>
> 2. **Code LLaMA results** - Thanks for checking the details in our Appendix. Code models are indeed more robust in tool using, CodeLLaMA is not an exception. CodeLLaMA 34b is even better than LLaMA2-70b on most of the tasks. We attached the results below, please compare them with the other results in Appendix Table 8, we will add them in the paper in the next version.
>
> | Model | Open weather | Cat API | Home Search | Trip Booking | Google Sheet | Virtual Home | WebShop (long) | WebShop (short) | TableTop |
> | ----------------------------- | ----------- | ----------- | --------- | ---------- | ----------- | ---------------------- | -------- | ---------- | ----------- |
> | CodeLlama-34b-hf | 96.39 | 85.0 | 88.0 | 88.33 | 64.29 | 34.0 / 24.65 | 0.0 | 5.53 | 51.32 |
> | CodeLlama-34b-Instruct-hf | 94.28 | 86.0 | 90.0 | 89.17 | 61.11 | 31.0 / 24.34 | 0.0 | 25.99 | 47.46 |
> | CodeLlama-34b-Python-hf | 91.11 | 88.42 | 91.0 | 85.83 | 55.87 | 28.0 / 21.24 | 0.0 | 6.47 | 33.33 |
> | CodeLlama-13b-hf | 96.0 | 86.83 | 87.0 | 70.83 | 51.26 | 35.0 / 22.28 | 0.0 | 0.0 | 40.95 |
> | CodeLlama-13b-Instruct-hf | 98.0 | 89.58 | 85.0 | 72.5 | 48.97 | 31.0 / 22.56 | 0.0 | 9.7 | 57.62 |
> | CodeLlama-13b-Python-hf | 93.0 | 92.35 | 86.0 | 62.5 | 50.79 | 37.0 / 21.94 | 0.0 | 2.4 | 41.53 |
> | CodeLlama-7b-hf | 86.0 | 92.0 | 74.0 | 63.33 | 38.08 | 35.0 / 21.97 | 0.0 | 0.0 | 11.16 |
> | CodeLlama-7b-Instruct-hf | 90.0 | 94.0 | 78.0 | 61.67 | 41.27 | 32.0 / 21.95 | 0.0 | 0.0 | 16.98 |
> | CodeLlama-7b-Python-hf | 83.0 | 88.0 | 83.0 | 68.33 | 49.13 | 31.0 / 21.33 | 0.0 | 1.58 | 22.86 |
>
> 3. **About Google Sheet Columns** - We present the meta data in markdown format to the model in prompt during inference. So the model always has access to all the column names as well as the rows of the content.

---

> > ### Author Response · Authors · 2023-11-22
> >
> > Dear Reviewer 3VL8,
> >
> > We sincerely appreciate your valuable and constructive feedback. We would like to kindly remind you that the author/reviewer discussion phase concludes on November 22nd. We hope our responses and additional experiments have addressed your concerns and improved the paper’s quality. If you have any further suggestions or comments, please feel free to share them. We are looking forward to a constructive discussion during the rebuttal phase.
> >
> > Best Regards,
> >
> > The Authors

---

### Official Review · Reviewer_dbqH · 2023-11-03

**Soundness:** 2 fair
**Presentation:** 2 fair
**Contribution:** 1 poor
**Rating:** 5
**Confidence:** 3

**Summary:**

The paper investigates how open-source LLMs can be improved to match the tool manipulation capabilities of proprietary, closed-API models like OpenAI's GPT-4. The authors identify three main challenges faced by open-source LLMs: incorrect API naming, improper argument values, and non-executable outputs. To address these issues, authors experiment with model alignment via programmatic data generation, in-context demonstration with retrieval, and system prompts, and adapt them for tool manipulation. They evaluate these methods using ToolBench, a benchmark they created consisting of various software tools, and find that their approaches can significantly enhance open-source LLM performance.

**Strengths:**

* New Benchmark: The creation and utilization of the new ToolBench benchmark for a diverse range of software tools allow for a thorough and systematic evaluation of the tool manipulation capabilities of various LLMs. This provides a soliid foundation for comparison and progress tracking in future research.
* Focus on open source models: The paper successfully applies novel techniques like model alignment, system prompts, and retrieval augmented in-context demonstration to enhance the capabilities of open-source LLMs.

**Weaknesses:**

Scalability concerns: the paper does not adequately adderess how the necessary amount of human supervision would scale with the increasing complexity of tasks or with different types of tasks beyond those tested.

**Questions:**

Can you add ReAct style prompting to the set of baselines?

In Section 4, how do you ensure that the system prompt used to regulate generation does not suppress the creativity of language models, which can be crucial for generating novel solutions to programming problems? Could this constraint inadvertently limit the model’s ability to generate innovative API usages that have not been programmed into its training data?

In section 5.1, you note that the benchmark involves tasks that require a different number of API functions, ranging from 2 to 108, to achieve the goal. Could you provide some performance metrics that would correlate with the number of API functions required to fulfill each goal?

---

> ### Author Response · Authors · 2023-11-15
>
> Thanks for recognizing the value of our work as well as your constructive feedback!
>
> **Regarding the strengths and motivation**
>
> Rather than completely solving the tool using issues for LLMs, our main motivation of this paper is more like opening the door for the community to explore and improve models/algorithms on tool using. As you mentioned in the strengths section, we sincerely hope our work can be published at ICLR so that the researchers and practitioners can 1) see and use our *ToolBench* to guide and evaluate their further research, 2) take the techniques we revisited as teasers to encourage further innovation that can fundamentally improve model tool using capability, and 3) exploit those techniques to enhance model performance on their own tools and tasks in practice.
>
> **Regarding the weaknesses and your questions**
>
> 1. **Scalability concerns** - We mainly revisit those three techniques in our work as proof-of-concepts to show that they are effective to improve model tool using capabilities with limited human supervision. But we will try to explain below a bit about human supervision scalability along each of the techniques we revisited.
>
> - About finetuning, we wanted to keep involving as minimum human supervision as possible. With the training data details in appendix section C.1, the dataset is just prepared by 1 author of us in just one day using the programmatic way described in section 4.1. Although we don’t have a scaling law study around it, the small collected dataset can already boost model performance by a lot, see Appendix Table 11 for the performance breakdown. We leave the scaling law study as well as the transferability across tasks as future work, where we would work with the community together to push it further.
>
> - About example retrieval, we showed the scaling law of it in Figure 4 on one of the *Toolbench* tasks. We found similar trend for other tasks as well, but only showed a typical one for space limit. In practice, for each given tool or a set of API functions, we found it is usually easy to collect a bunch of demonstration examples for most of the typical uses cases. Further, in the experiment for Figure 4, we made sure that the test cases all require completely different API function combinations to achieve each goal than the ones in the 10 demonstration examples, indicating that the models can indeed generalize to unseen API combinations from limited retrieved examples. Thus, this quick accuracy saturation trend is pretty satisfying in practice.
>
> - About the system prompt, it is a one time job that can be shared across multiple tasks. We provided more details about it in point 3 below and Appendix section B.1.
>
> 2. **ReAct style prompting** - We internally explored a bit about different prompting methods following CoT[1], ReAct[2], and Reflexion[3], but they do not trivially work out-of-the-box and behaves much worse than the simple few-shot retrieval augmented generation we mentioned in the paper. But we agree there are a lot more to explore along this direction, for example the new work Toolchain [4], where they compared their method against ReAct[2] and AdaPlanner[5], on *ToolBench*. We are more than excited to join the community to explore new algorithms to further improve the model performance on tool using and their performance on *ToolBench*.
>
> 3. **System prompt** - The system prompt we used are provided in Appendix B.1. It is a pretty general prompt and is not tied to any specific task. In our experiments, both the API definition and demonstration examples are retrieved from the database for each query and plugged into the prompt during inference. So we were targeting to have a general system prompt that can generalize to different tasks. Further, as listed in Appendix Table 11 for the breakdown, this system prompt alone can improve the model quality over the zero-shot baseline, especially on the easier tasks.
>
> 4. **Performance v.s. Number of API functions** - Usually, the difficulty of a given task is not only correlated with the number of API functions, but also with the difference between the demonstration examples provided and the real test cases. Thus, we proposed a new metric “API complexity” in section 5.2 to quantify this challenge. We also provided the details of this metrics as well as the model performance against tasks with different “API complexity” in appendix section D.
>
>
> [1] Wang et al, Plan-and-Solve Prompting: Improving Zero-Shot Chain-of-Thought Reasoning by Large Language Models
>
> [2] Yao et al, ReAct: Synergizing Reasoning and Acting in Language Models
>
> [3] Shinn et al, Reflexion: Language Agents with Verbal Reinforcement Learning
>
> [4] Zhuang et al, ToolChain*: Efficient Action Space Navigation in Large Language Models with A* Search
>
> [5] Sun et al, AdaPlanner: Adaptive Planning from Feedback with Language Models

---

> ### Author Response · Authors · 2023-11-22
>
> Dear Reviewer dbqH,
>
> We sincerely appreciate your valuable and constructive feedback. We would like to kindly remind you that the author/reviewer discussion phase concludes on November 22nd. We hope our responses have addressed your concerns and improved the paper’s quality. If you have any further suggestions or comments, please feel free to share them. We are looking forward to a constructive discussion during the rebuttal phase.
>
> Best Regards,
>
> The Authors

---

### Meta-Review · Area_Chair_hpV8 · 2023-12-08

**Metareview:**

The paper explores improvement to open-source LLMs to match the tool manipulation abilities of proprietary models. It identifies three primary issues in open-source LLMs: inaccurate API naming, unsuitable argument values, and outputs that can't be executed. The authors tackle these issues through techniques like model alignment, demonstrations coupled with retrieval methods, and tailored system prompts for tool manipulation. They evaluate these strategies with ToolBench, their custom benchmark comprising various tools.

While the newly proposed ToolBench dataset serves as a solid basis for comparison and progress tracking in the field, many reviewers have expressed concerns regarding its differentiation from other similar benchmarks like GPT4Tool, Gorilla, ToolQA, and others. After thoroughly reviewing the rebuttal and the subsequent discussion from the reviewers, I find that these concerns remain valid. Consequently, I recommend a rejection rating.

**Justification For Why Not Higher Score:**

As mentioned above, many reviewers have expressed concerns regarding its differentiation from other similar benchmarks like GPT4Tool, Gorilla, ToolQA, and others. After thoroughly reviewing the rebuttal and the subsequent discussion from the reviewers, I find that these concerns remain valid.

**Justification For Why Not Lower Score:**

N/A

---

### Decision · Program_Chairs · 2024-01-16

Reject